# Predictive State Observer-Based Aircraft Distributed Formation Tracking Considering Input Delay and Saturations

Liguo Sun, Xiaoyu Liu, Wenqian Tan *, Yi Deng, Junkai Jiao and Mengjie Zhao

School of Aeronautic Science and Engineering, Beihang University, Beijing 100191, China;
l.g.sun@buaa.edu.cn (L.S.) ; liuxiaoyu2016@buaa.edu.cn (X.L.); deng_yi@buaa.edu.cn (Y.D.);
jjk_mail@buaa.edu.cn (J.J.); sy2205415@buaa.edu.cn (M.Z.)
* Correspondence: tanwenqian@buaa.edu.cn; Tel.: +86-188-1074-6402

**Abstract:** This paper investigates a fully distributed time-varying formation tracking problem for a group of fixed-wing aircraft. The fixed-wing aircraft formation control system consists of an outer-loop trajectory control subsystem and an inner-loop attitude control subsystem. For fixed-wing aircraft, it is crucial to consider the time delay of the engine response, the model uncertainties, the tracking capability of the attitude commands in the inner loop, and other agility performances of the aircraft. To address the problems related to the input time delay and model uncertainties, a predictive extended state observer-based fully distributed time-varying formation tracking control (PESO-TVFTC) protocol is proposed. To satisfy the constraints set by the attitude tracking quickness and the trajectory tracking smoothness, the low gain feedback technique is introduced in the protocol to keep the control inputs for the outer loop within the desired saturation constraints. Through theoretical analysis, it is proved that the multiple aircraft systems can achieve time-varying formation tracking consensus under specific initial conditions and feasibility conditions, and it is shown that the upper bounds of the PESO gains are restricted by the time delay. Numerical simulations are used to demonstrate the effectiveness of and the improvements in the proposed method.

**Keywords:** formation tracking control; input time delay; input saturation; consensus method; distributed control

## 1. Introduction

With the development of unmanned aerial vehicle (UAV) technology and increasing demand for mission requirements [1–4], aircraft cooperative formation control has received considerable attention. In the past few decades, researchers have proposed many methods for cooperative formation including leader–follower, virtual structure, behavior-based methods, which are applied in the formation control of UAVs [1,2], spacecrafts [5,6], robotics [7,8], etc. Compared with the above three methods, the consensus method [9] is more robust and extensible, and is a general framework that contains these methods [10]. Thus, the consensus control method has been a research topic for more than 10 years, and has been applied in the time-varying formation [11], formation tracking [12,13], obstacle avoidance formation flight of UAV [14], etc. However, the consensus methods in [11–14] are not fully distributed.

The fully distributed consensus control protocol is crucial to the improvement of the robustness and the extensibility of the system. Full distribution means that each agent in the formation only uses the state information or output information of itself or its neighbors communicating with it, and requires no global information. As a result, the number of transmitted signals is reduced, and the robustness and the extensibility of the system are improved. However, in [11–14], the Laplacian matrix, which is global information, is used for consensus protocol design, which means that the protocols are not fully distributed. To solve the above problems, Li proposed a fully distributed consensus protocol for a general linear system and Lipschitz nonlinear system under undirected

communication topology [15] and then extended it to a linear multi-agent system (MAS) with directed communication topology [16]. Based on the above framework, Chen designed a fully distributed controller in the case of actuators' failure, realizing the exponential output consensus for heterogeneous MAS [17]. Jiang designed a controller containing a fully distributed adaptive observer and Luenberger observer, addressing the time-varying formation containment problem for a heterogeneous linear MAS [18]. Zhang designed a fully distributed time-varying formation tracking control protocol based on a finite time convergent extended state observer, addressing the anti-disturbance formation tracking problem for a quadrotor formation [19]. Cheng designed a distributed adaptive states and output feedback protocol, solving the formation control problem of a linear MAS under event-triggered communications constraints [20].

In reality, time delays are inevitable in fixed-wing aircraft, such as the input time delay generated by actuators and engines or the time delay caused by information transmission [21,22]. These delays slow down the response of the aircraft, thus degrading the rapidity and accuracy of the formation tracking, or even causing state divergence of the aircraft. However, in studies of consensus-based methods for the formation flight of fixed-wing aircraft [14,23,24], the time-delay problem has scarcely been investigated. Therefore, it is necessary to investigate the time-delay problem when designing the time-varying formation tracking consensus protocol for fixed-wing aircraft. The reported solutions to the time-delay problem can be classified into the robust analysis manner and the active compensation manner. The robust analysis manner is to seek a controller parameter range or a maximal allowable time delay that guarantees the stability of the formation system. For example, Wang studied an MAS with a steady communication delay, giving the upper bound of the time delay that could stabilize the system, and showed that the upper bound was related to the dynamics of the system and the topology of the communication network [25]. Zong selected the appropriate control gain according to the time delay and noise intensities in the measurement term, achieving stochastic consensus of the continuous-time MAS [26]. Zhang further used absolute velocity and relative position measurements, achieving consensus control of a continuous-time second-order MAS with time delay and multiplicative noise by choosing appropriate control gains [27]. Nevertheless, the solution in [25] can only tolerate relatively short time delays, and global information is needed in [26,27]. Another solution is the active compensation manner [28], which compensates for the time delay in the controller actively. One of the most popular compensation methods is to predict the values of states by a predictor, and to use the predictive values in the design of the controller. Jiang used Arstein's model reduction technique [29] to design a state predictor to convert an MAS with an input delay into a delay-free one [30]. Zhou used a truncated predictor feedback approach to solve the consensus problem of a high-order MAS with input and communication delays [31]. Wang designed a cascade structure predictive observer, achieving consensus control of an MAS with long input delays [32]. However, for predictor-based approaches, exact dynamic information of the system is required, which is difficult to obtain in reality due to the unknown dynamics and internal/external disturbances. To solve the above problems, the combination of the predictor and the extended state observer (ESO) in the active disturbance rejection control (ADRC) [33] is an effective strategy. For the consensus problem of an MAS with input and output time delay under external disturbances, Wang designed a predictor-based ESO [34]. Considering input time delays and disturbances, Jiang designed an adaptive predictive ESO to achieve the fully distributed leader–follower consensus of the linear MAS with an unknown leader [35]. However, it should be noted that in [34,35], the delay time is constant and it is the consensus tracking problem that is solved. In addition, the control protocol in [34] is not fully distributed. For fixed-wing aircraft with uncertainties and time delays, how to design a fully distributed time-varying formation tracking control protocol is still open.

Apart from the time delay, in the actual system, there exist many constraints such as input constraints, state constraints, and output constraints [14,36,37]. For the aircraft system, considering the attitude tracking quickness, the trajectory tracking smoothness,

and the feasibility of the attitude commands, the input saturation is the constraint that cannot be ignored. The low gain feedback (LGF) technology [38] is an efficient solution to the input saturation problem of the formation system. Su used the LGF method to solve the input saturation constraints in the consensus control of a linear MAS [39]. Chu designed the observer-based fully distributed control protocol by using the LGF method, achieving consensus tracking of a linear MAS. For a heterogeneous MAS [40], Yang used the LGF method to eliminate the saturation effect of the actuator, achieving its output consensus [41]. Su further extended the LGF method to the design of the output consensus protocol for the discrete system [42]. Wang proposed a metamorphic LGF technology to design the consensus tracking control protocol for an uncertain MAS with input saturation constraints [43] and extended the result to the protocol design for an MAS with multiple saturation levels and switching communication topologies [44]. However, for multiple fixed-wing aircraft with input time delay and saturation constraints, how to design the formation control protocol with the LGF method while solving the input delay problem is still open.

Based on the discussion above, to satisfy the performance constraints of the aircraft engines, the input time-delay problem needs to be addressed. Simultaneously, considering the attitude tracking quickness, the trajectory tracking smoothness, and the feasibility of the attitude commands, the input saturation problem also needs to be solved.

In this paper, we aim to address the fully distributed time-varying formation tracking control of fixed-wing aircraft with the simultaneous presence of uncertainties, input time delay, and input saturation. Inspired by the literature and mainly motivated by the work in [45], a predictive extended state observer-based fully distributed time-varying formation tracking control (PESO-TVFTC) protocol designed using the low gain feedback (LGF) technology is proposed. Then, the stability of the system is proved through theoretical analysis, and the effectiveness and improvement of the proposed method are demonstrated using numerical simulation. The main contributions of this paper are three-fold: Firstly, the PESO-TVFTC protocol is proposed in this paper, which is suitable for time-varying formation tracking control of a multiple aircraft system with uncertainties and time-varying input time delay. Secondly, the input saturation constraints and input time delay are both considered simultaneously, and the LGF technology is used to design the proposed PESO-TVFTC protocol. Thirdly, the proposed PESO-TVFTC protocol is fully distributed, requiring no global information.

It is noted that the consensus control problem of a multi-agent system with input time delay has also been studied in [30–32,34,35]. However, the work in this paper is different to theirs. Firstly, compared to [34,35], it is the time-varying formation tracking problem rather than the classical leader–follower consensus problem that is studied, and compared to [34], the protocol proposed in this paper is fully distributed. Secondly, compared to [30,32,35], the system studied in this paper can be nonlinear and has uncertainties, and the input delay can be time-varying. Lastly, different from [30–32,34,35], this paper not only solves the input delay problem but also considers the input saturation constraints and embeds the low-gain feedback technology into the control protocol.

The remainder of this paper is organized as follows. Mathematical preliminaries, definitions, lemmas, and problem formulation are given in Section 2. In Section 3, the PESO-TVFTC protocol is proposed and designed using the LGF method, and the theoretical analysis is given. In Section 4, the numerical simulation of the formation assembly and the formation change of the fixed-wing aircraft flying through a valley slit is carried out to demonstrate the effectiveness of and improvements in the proposed method. Finally, Section 5 concludes this paper.

## 2. Preliminaries and Problem Statement

### 2.1. Basic Concepts on Graph Theory

A directed graph $G = \{V, E, W\}$ is defined to describe the interaction topology of the multi-aircraft system. The weighted directed graph $G$ consists of a set of nodes

$V = \{v_1, v_2, \ldots, v_N\}$, a set of edges $E \subseteq \{(v_i, v_j) : v_i, v_j \in V\}$, and a weighted adjacency $W = [w_{ij}] \in \mathbb{R}^2$ with nonnegative elements $w_{ij}$. The edge of $G$ is denoted by $v_{ij} = (v_i, v_j)$, where the node $v_i$ is called a neighbor of the node $v_j$. The entries in $W$ are defined in that $w_{ji} > 0$ if and only if $v_{ij} \in E$, otherwise, $w_{ji} = 0$ for all $i, j \in \{1, 2, \ldots, N\}$. In addition, $w_{ii} = 0$ for all $i \in \{1, 2, \ldots, N\}$. $Q_i = \{v_j \in V : v_{ji} \in E\}$ denotes the neighbor set of the node $v_i$. Let $\deg_{in}(v_i) = \sum_{j=1}^{N} w_{ij}$ be the in-degree of the node $v_i$ and $D = diag\{\deg_{in}(v_i), i = 1, 2, \ldots, N\}$ be the degree matrix of $G$. Then, define the Laplacian matrix $L \in \mathbb{R}^{N \times N}$ of $G$ as $L = D - W$. If there is a directed path from one node, which is called the root to every other node, the directed graph is deemed to have a spanning tree.

*2.2. Definitions and Lemmas*

The leader and follower are defined as follows.

**Definition 1** ([16]). *An aircraft is called a leader if its corresponding node in the directed graph does not have the incoming edge and is called a follower if it has at least one incoming edge.*

Considering the actual formation flying of multiple fixed-wing aircraft. There is a leader in the formation tracking the specific reference trajectory signals, which plays a leading role in the multiple aircraft systems. The other $N - 1$ aircraft are followers, who follow the trajectory of the leader and form a specific formation. Let $N$ be the subscript of the leader and $\bar{F} = \{1, 2, \ldots, N - 1\}$ be the subscript set of followers.

**Assumption 1** ([19]). *The directed graph $G$ contains a spanning tree with the leader as the root node.*

**Remark 1.** *$G$ containing a spanning tree means that there exists at least a path from the leader to the follower i. In reality, this means that each follower can directly or indirectly receive information from the leader, which is necessary for formation. The same assumption can be found in [12,19].*

In order to express the communication topology of the aircraft formation, the Laplacian matrix is introduced. Then, the following lemmas are satisfied from Assumption 1, which is subsequently used to prove the stability of the formation tracking system.

**Lemma 1** ([9]). *If $L$ is the Laplacian matrix of a directed interaction topology with a spanning tree, zero is a simple eigenvalue of $L$ associated with the eigenvector 1, and all the other nonzero eigenvalues are located in the right-half plane of the imaginary axis.*

From Definition 1, one can obtain the Laplacian matrix $L$ corresponding to the directed graph $G$ is as follows:

$$L = \begin{bmatrix} L_1 & L_2 \\ \mathbf{0}_{1 \times (N-1)} & 0 \end{bmatrix} \tag{1}$$

where $L_1 \in \mathbb{R}^{(N-1) \times (N-1)}$ and $L_2 \in \mathbb{R}^{(N-1) \times 1}$. From Lemma 1 and Assumption 1, one can find that all the eigenvalues of $L_1$ have positive real parts. Then, one can obtain that $L_1$ is nonsingular and is a diagonally dominant $M$-matrix.

**Lemma 2** ([16]). *For a diagonally dominant M-matrix, there exists a positive diagonal matrix $R = diag\{r_1, r_2, \ldots, r_{N-1}\}$ such that $RL_1 + L_1^T R \geq \lambda_0 I_{N-1}$, where $\lambda_0$ denotes the smallest eigenvalues of $RL_1 + L_1^T R$ and $\bar{r} = [r_1, r_2, \ldots, r_{N-1}]^T = (L_1^T)^{-1} \mathbf{1}_{N-1}$.*

**Lemma 3.** *(Young's inequality) If p and q are nonnegative real numbers and m and n are positive real numbers satisfying $(1/m) + (1/n) = 1$, then $pq \leq (p^m/m) + (q^n/n)$.*

### 3. Mathematical Model of Fixed-Wing Aircraft

Before giving the mathematical model of fixed-wing aircraft, the axes used in this paper are defined.

**Definition 2.** *Letting the initial position of the leader be the origin, the directions of the x, y, and z axes are parallel to the $x_k$, $y_k$ and $z_k$ directions of the flight path coordinate system of the ith aircraft, respectively, where $i \in \bar{F}$.*

The dynamics of the outer-loop subsystem of the aircraft are shown as follows:

$$\begin{cases} \dot{p}_{i,j}^k(t) = v_{i,j}^k(t) \\ \dot{v}_{i,j}^k(t) = \bar{u}_{i,j}^k(t - \tau(t)) + \bar{d}_{i,j}^k(t) \\ \bar{u}_{i,j}^k(t - \tau(t)) = \text{sat}(u_{i,j}^k(t - \tau(t))) \end{cases} \tag{2}$$

where $i \in \{1, 2, \ldots, N\}, k \in \bar{F}, j \in \{x, y, z\}$. Superscript $k$ indicates that the reference coordinate system is defined by Definition 2 regarding the $k$th aircraft. $p_{i,j}^k(t), v_{i,j}^k(t), u_{i,j}^k(t - \tau(t))$, and $\bar{d}_{i,j}^k(t)$ represent the positions, velocities, control inputs of the outer-loop subsystem, and synthetic uncertainties of the $i$th aircraft in the $j$ direction at time $t$, respectively, where the control inputs satisfy $u_{i,j}^k(t) = 0$ for $t < 0$, and synthetic uncertainties include external disturbances and unknown internal dynamics. $\tau(t)$ represents an input time delay. $\text{sat}(\cdot) : \mathbb{R} \to \mathbb{R}$ is a saturation function defined as $\text{sat}(\omega) = \text{sgn}(\omega) \cdot \min\{|\omega|, \bar{M}_j\}$. $\bar{M}_j > 0$ are input saturation constraints, which are mainly determined by the thrust-to-weight ratio, the lift–drag ratio, the structural strength, and the maximum overload of the aircraft the pilot can withstand. It should be noted that, unless specified, the input in the remainder of this paper refers to the control input of the outer-loop subsystem.

The longitudinal and lateral dynamics of the inner-loop subsystem of the $i$th aircraft are shown as follows:

$$\dot{\bar{s}}_{lon} = A_{lon}\bar{s}_{lon} + B_{lon}\delta_{lon} \tag{3}$$

$$\dot{\bar{s}}_{lat} = A_{lat}\bar{s}_{lat} + B_{lat}\delta_{lat} \tag{4}$$

with $\bar{s}_{lon} = [V, \alpha, \theta, q]^T, \bar{s}_{lat} = [\beta, p, r, \phi, \psi, \chi, \mu]^T, \delta_{lon} = [\delta_e, \delta_T]^T, \delta_{lon} = [\delta_a, \delta_r]^T$, where $A_{lon} \in \mathbb{R}^{4 \times 4}, B_{lon} \in \mathbb{R}^{4 \times 2}, A_{lat} \in \mathbb{R}^{7 \times 7}, B_{lat} \in \mathbb{R}^{7 \times 2}$. $V$ is the airspeed of the aircraft. $\phi, \theta, \psi, \mu, \alpha, \beta, \chi$ are the roll angle, pitch angle, yaw angle, back angle, attack angle, side-slip angle, and heading angle, respectively. $p, q, r$ are the angular rates of roll, pitch, and yaw, respectively. $\delta_T, \delta_a, \delta_e, \delta_r$ are the control inputs, namely, the engine power level angle and the deflections of the aileron, elevator, and rudder, respectively.

The transformation between the outer-loop inputs and inner-loop commands of the $i$th aircraft is derived as follows:

$$\begin{cases} T\cos\alpha - mg\sin\gamma - F_D = m\bar{u}_{i,x}^i \\ T\sin\alpha\sin\mu + F_L\sin\mu = m\bar{u}_{i,y}^i \\ -T\sin\alpha\cos\mu + mg\cos\gamma - F_L\cos\mu = m\bar{u}_{i,z}^i \end{cases} \tag{5}$$

with

$$\begin{aligned} F_L &= 1/2\rho V^2 S(C_{L_0} + C_{L_\alpha}\Delta\alpha) \\ F_D &= 1/2\rho V^2 S(C_{D_0} + C_{D_\alpha}\Delta\alpha) \\ T &= T_0 + T_{\delta_T}\delta_T \\ \alpha &= \alpha_0 + \Delta\alpha \end{aligned}$$

where $i \in \bar{F}$ and $\rho, S$ represent the current atmospheric density and wing surface area, respectively, $\gamma$ represents the flight path angle and has $\gamma = \theta - \alpha$, $C_{L_0}, C_{D_0}$ are the current lift and drag coefficient, respectively, $C_{L_\alpha}, C_{D_\alpha}$ represent the aerodynamic derivatives, $T_0$ and $\alpha_0$ represent the current thrust and attack angle, $T_{\delta_T}$ represents the thrust generated by per unit $\delta_T$, and $m, g$ represent the mass of the aircraft and gravitational acceleration,

respectively. As shown in the following Figure 1, by using Equation (5), one can solve the commands of $\delta_T, \mu, \Delta\alpha$ from $\bar{u}_{i,x}^i, \bar{u}_{i,y}^i, \bar{u}_{i,z}^i$.

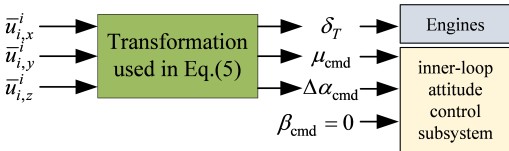

**Figure 1.** The inputs and outputs of the transformation used in Equation (5).

The dynamics of fixed-wing aircraft satisfy the following assumption:

**Assumption 2** ([46]). *The $u_{N,j}^k$ and $\dot{u}_{N,j}^k$ of the leader are bounded and are unknown to followers.*

**Remark 2.** *Due to the constraints of the power and the agility performance of the aircraft, the thrust, the deflection of the actuator, and their rate of change are limited, so the input $u_{N,j}^k$ and $\dot{u}_{N,j}^k$ are bounded. Thus, Assumption 2 is reasonable.*

**Remark 3.** *The fixed-wing aircraft system consists of an outer-loop subsystem (2) and an inner-loop subsystem (3) and (4). The trajectories are controlled in the outer-loop subsystem, and the attitudes are controlled in the inner-loop subsystem. The formation tracking control framework is shown in Figure 2. To achieve the formation tracking consensus of multiple aircraft systems, this paper mainly focuses on the control protocol design in the outer-loop subsystem.*

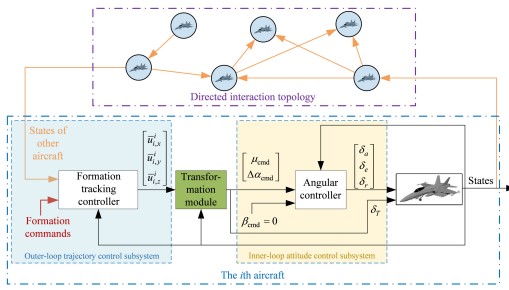

**Figure 2.** Formation tracking control framework for fixed-wing aircraft.

*Problem Formulation*

Let the piecewise continuously differentiable vector $\boldsymbol{h}_{i,j}(t) = [h_{pi,j}{}^T(t), h_{vi,j}{}^T(t)]^T \in \mathbb{R}^2$, $i \in \bar{F}, j \in \{x, y, z\}$ be the time-varying formation command vector of the $i$th follower in the $j$ direction, whose physical meaning represents offset values of position or velocity from the leader aircraft. After that, the formation command vector of the multiple aircraft systems can be expressed as $\boldsymbol{h}_j(t) = [\boldsymbol{h}_{1,j}{}^T, \boldsymbol{h}_{2,j}{}^T, \ldots, \boldsymbol{h}_{N-1,j}{}^T]^T$. Then, the states of $i$th aircraft in the $j$ direction are denoted by $\boldsymbol{s}_{i,j}^k(t) = [p_{i,j}^k, v_{i,j}^k]^T, i = 1, 2, \ldots, N$.

**Assumption 3.** *The formation command vectors $\boldsymbol{h}_{i,j}(t)$ and $\dot{\boldsymbol{h}}_{vi,j}(t), i \in \bar{F}, j \in \{x, y, z\}$ are bounded with respect to $t$.*

**Remark 4.** *Assumption 3 is commonly used in the control field and is used to prevent the divergence caused by excessive commands.*

Then, the time-varying formation tracking problem is defined as follows:

**Definition 3.** *The multiple aircraft systems* (2)–(4) *are said to achieve time-varying formation tracking consensus if for any a priori given bounded set* $\mathbf{\Omega}_0 \in \mathbb{R}^2$ *and the initial states satisfying* $s_{i,j}^k(0) \in \mathbf{\Omega}_0, i \in \{1,2,\ldots,N\}, k \in \bar{F}, j \in \{x,y,z\}$, *and the following formula holds*

$$
\begin{aligned}
&\lim_{t \to \infty} (s_{i,j}^k(t) - h_{i,j}(t) - s_{N,j}^k(t)) = O(\varepsilon), \\
&i \in \bar{F}, k = i, j \in \{x,y,z\}
\end{aligned}
\tag{6}
$$

*where* $\varepsilon$ *is a sufficiently small positive constant.*

## 4. PESO-TVFTC Protocol Design and Analysis

In this section, the PESO is introduced into the TVFTC to deal with the time delay issue associated with the engine thrust response. As a result, the PESO-TVFTC protocol is proposed. In order to circumvent the input saturation problem associated with the trajectory control outer loop, the low gain feedback method is adopted in the design of the PESO-TVFTC protocol. Then, the time-varying formation tracking consensus of the multiple aircraft systems with time delay and saturation achieved by the PESO-TVFTC protocol is proved.

### 4.1. PESO-TVFTC Protocol Design

Considering the general form of systems with time-varying input delay and input saturation based on system (2):

$$
\begin{cases}
\dot{\bar{z}}_{i,k}^k(t) = f_0(t, s_{i,j}^k(t), \bar{z}_{i,j}^k(t), \bar{d}_{i,j}^k(t)) \\
\dot{s}_{i,j}^k(t) = \boldsymbol{A} s_{i,j}^k(t) + \boldsymbol{B}[f(t, s_{i,j}^k(t), \bar{z}_{i,j}^k(t), \bar{d}_{i,j}^k(t)) + \bar{u}_{i,j}^k(t - \tau(t))] \\
\bar{y}_{i,j}^k(t) = \boldsymbol{C} s_{i,j}^k(t) \\
\bar{u}_{i,j}^k(t - \tau(t)) = \mathrm{sat}(u_{i,j}^k(t - \tau(t)))
\end{cases}
\tag{7}
$$

where $i \in \{1,2,\ldots,N\}, k \in \bar{F}, j \in \{x,y,z\}, t \geq 0, s_{i,j}^k(t) = [p_{i,j}^k, v_{i,j}^k]^T$, and $\bar{z}_{i,j}^k(t) \in \mathbb{R}^p$ represent states [45], $\bar{y}_{i,j}^k(t) \in \mathbb{R}$ represent measured outputs, and $\bar{d}_{i,j}^k(t) \in \mathbb{R}$ are external disturbances. $f_0(\cdot) : \mathbb{R}^+ \times \mathbb{R}^n \times \mathbb{R}^p \times \mathbb{R} \to \mathbb{R}^p$ and $f(\cdot) : \mathbb{R}^+ \times \mathbb{R}^n \times \mathbb{R}^p \times \mathbb{R} \to \mathbb{R}$ are unknown continuously differentiable functions, and the triple $(A, B, C)$ represents a chain of integrators satisfying asymptotically null controllable with bounded controls (ANCBC), i.e.,

$$
\boldsymbol{A} = \begin{bmatrix} 0 & 1 \\ 0 & 0 \end{bmatrix}, \boldsymbol{B} = \begin{bmatrix} 0 \\ 1 \end{bmatrix}, \boldsymbol{C} = \begin{bmatrix} 1 & 0 \end{bmatrix}.
$$

In this paper, the systems (7) with time-varying input delay satisfy the following assumption:

**Assumption 4** ([47]). *The time delay* $\tau(t)$ *is continuously differentiable and satisfies* $0 \leq \tau(t) \leq \bar{h}$ *and* $\dot{\tau}(t) < 1$, *where* $\bar{h} \in \mathbb{R}^+$ *represents the maximum input delay. In addition, the systems* (7) *do not escape to infinity when* $\tau(t) \in [0, \bar{h}]$.

**Assumption 5.** *The external disturbances* $\bar{d}_{i,j}^k(t)$ *and their derivative* $\dot{\bar{d}}_{i,j}^k(t), i, k \in \{1,2,\ldots,N\}, j \in \{x,y,z\}$ *are bounded.*

**Assumption 6.** *For* $\forall (t, s_{i,j}^k, \bar{z}_{i,j}^k, \bar{d}_{i,j}^k) \in \mathbb{R}^+ \times \mathbb{R}^n \times \mathbb{R}^p \times \mathbb{R}, i, k \in \{1,2,\ldots,N\}, j \in \{x,y,z\}$, *there exists a continuous function* $\bar{\psi}(\cdot) : \mathbb{R}^n \times \mathbb{R}^p \times \mathbb{R} \to \mathbb{R}^+$ *such that*

$$
\begin{aligned}
&\max\{ \left| f(t, s_{i,j}^k, \bar{z}_{i,j}^k, \bar{d}_{i,j}^k) \right|, \left| f_0(t, s_{i,j}^k, \bar{z}_{i,j}^k, \bar{d}_{i,j}^k) \right|, \\
&\left\| \nabla f(t, s_{i,j}^k, \bar{z}_{i,j}^k, \bar{d}_{i,j}^k) \right\|, \left\| \nabla f_0(t, s_{i,j}^k, \bar{z}_{i,j}^k, \bar{d}_{i,j}^k) \right\| \} \\
&\leq \bar{\psi}(s_{i,j}^k, \bar{z}_{i,j}^k, \bar{d}_{i,j}^k).
\end{aligned}
$$

**Assumption 7.** *For* $\forall (t, s_{i,j}^k, \bar{d}_{i,j}^k) \in \mathbb{R}^+ \times \mathbb{R}^n \times \mathbb{R}, i, k \in \{1, 2, \ldots, N\}, j \in \{x, y, z\}$, *there exists a positive definite function* $\bar{V}_0(\cdot) : \mathbb{R}^p \to \mathbb{R}^+$ *such that*

$$\frac{\partial \bar{V}_0(\bar{z}_{i,j}^k)}{\partial \bar{z}_{i,j}^k} f_0(t, s_{i,j}^k, \bar{z}_{i,j}^k, \bar{d}_{i,j}^k) \leq 0,$$
$$\forall \bar{z}_{i,j}^k : \left\| \bar{z}_{i,j}^k \right\| \geq \bar{l}(\left\| (s_{i,j}^k, \bar{d}_{i,j}^k) \right\|)$$

*where* $\bar{l}(\cdot)$ *is a class* $K_\infty$ *function.*

**Remark 5.** *Although delays exist in the response of the aircraft engine and the actuators, delays and their rate of change can not be infinite to ensure the maneuverability of the aircraft. Thus, Assumption 4 is reasonable. Moreover, since the energy of the external disturbance, such as gust, is finite naturally, Assumption 5 is reasonable. Moreover, according to Assumption 2, $u_{N,j}^k$ and $\dot{u}_{N,j}^k$ are both bounded, thus, the synthetic uncertainties are bounded and Assumptions 6 and 7 are reasonable.*

Define the delayed time point as $\varpi(t) = t - \tau(t)$. Under Assumption 4, $\varpi(t)$ is continuously differentiable and strictly increasing. Denote the inverse function of $\varpi(t)$ as $\vartheta(t) = \varpi^{-1}(t)$. It can be obtained that $\vartheta(t)$ is the prediction time point and $\vartheta(t) - t$ represents the prediction range. From Assumption 4, one can find that $\dot{\vartheta}(t) \in [\vartheta_1, \vartheta_2]$, where $\vartheta_1$ and $\vartheta_2$ are both positive constants. Then, one can design the following feedback control law such that the system (7) is stable:

$$\begin{aligned}
u_{i,j}^k(t) &= u_{0i,j}^k(s_{i,j}^k(\vartheta(t))) \\
&- f(\vartheta(t), s_{i,j}^k(\vartheta(t)), \bar{z}_{i,j}^k(\vartheta(t)), \bar{d}_{i,j}^k(\vartheta(t))),
\end{aligned} \tag{8}$$

where $u_{0i,j}^k(s_{i,j}^k(\vartheta(t)))$ represents the control law that can make the system stable after eliminating the nonlinear term, which needs to be designed. One can see that the control law shown in Equation (8) relies on the predictions of states and synthetic uncertainties at the time $\vartheta(t)$, which are unavailable. Thus, the PESO is designed to solve this problem.

Denote the extended states of the followers and leader as

$$\xi_{i,j}^k(\vartheta(t)) = f(\vartheta(t), s_{i,j}^k(\vartheta(t)), \bar{z}_{i,j}^k(\vartheta(t)), \bar{d}_{i,j}^k(\vartheta(t))),$$

$i \in \{1, 2, \ldots, N\}, k \in \bar{F}, j \in \{x, y, z\}$, and

$$\xi_{N,j}^k(\vartheta(t)) = f(\vartheta(t), s_{N,j}^k(\vartheta(t)), \bar{z}_{N,j}^k(\vartheta(t)), \bar{d}_{N,j}^k(\vartheta(t))) + u_{N,j}^k(t),$$

respectively. Then, the PESO in [45] is reformulated for the multiple aircraft systems as follows.

$$\begin{cases}
\dot{\hat{p}}_{i,j}^k(t) = \dot{\vartheta}(t)\hat{v}_{i,j}^k(t) + \dot{\vartheta}(t)\varepsilon g_1 [\bar{y}_{i,j}^k(t) - \hat{p}_{i,j}^k(\varpi(t))]/\varepsilon^2 \\
\dot{\hat{v}}_{i,j}^k(t) = \dot{\vartheta}(t)\hat{\xi}_{i,j}^k(t) + \dot{\vartheta}(t)g_2 [\bar{y}_{i,j}^k(t) - \hat{p}_{i,j}^k(\varpi(t))]/\varepsilon^2 \\
\quad + \dot{\vartheta}(t)u_{i,j}^k(t) \\
\dot{\hat{\xi}}_{i,j}^k(t) = \dot{\vartheta}(t)\varepsilon^{-1}g_3 [\bar{y}_{i,j}^k(t) - \hat{p}_{i,j}^k(\varpi(t))]/\varepsilon^2
\end{cases} \tag{9}$$

$$\begin{cases}
\dot{\hat{p}}_{N,j}^k(t) = \dot{\vartheta}(t)\hat{v}_{N,j}^k(t) + \dot{\vartheta}(t)\varepsilon g_1 [\bar{y}_{i,j}^k(t) - \hat{p}_{N,j}^k(\varpi(t))]/\varepsilon^2 \\
\dot{\hat{v}}_{N,j}^k(t) = \dot{\vartheta}(t)\hat{\xi}_{N,j}^k(t) + \dot{\vartheta}(t)g_2 [\bar{y}_{i,j}^k(t) - \hat{p}_{N,j}^k(\varpi(t))]/\varepsilon^2 \\
\dot{\hat{\xi}}_{N,j}^k(t) = \dot{\vartheta}(t)\varepsilon^{-1}g_3 [\bar{y}_{i,j}^k(t) - \hat{p}_{N,j}^k(\varpi(t))]/\varepsilon^2
\end{cases} \tag{10}$$

where $i, k \in \bar{F}, j \in \{x, y, z\}, \hat{s}_{i,j}^k(t) = [\hat{p}_{i,j}^k(t), \hat{v}_{i,j}^k(t), \hat{\xi}_{i,j}^k(t)]^T, i \in \{1, 2, \ldots, N\}$ is the observer state with the initial conditions $\hat{s}_{i,j}^k(t) = 0, \forall t \in [-\tau(0), 0]$. $g_i(\cdot) : \mathbb{R} \to \mathbb{R}, i = 1, 2, 3$ are

continuously differentiable functions that can be designed, and $\varepsilon$ is a sufficiently small positive constant. In Equations (9) and (10), the physical meanings of the extended states are the magnitude of the acceleration.

**Remark 6.** *Different from the ESO in [33,48], the input of the PESOs (9) and (10) is $(\bar{y}_{i,j}^k(t) - \hat{p}_{i,j}^k(\omega(t)))/\varepsilon^2$ rather than $(\bar{y}_{i,j}^k(t) - \hat{p}_{i,j}^k(t))/\varepsilon^2$, and $\hat{s}_{i,j}^k(t)$ is the estimation of $[p_{i,j}^k(\vartheta(t)), v_{i,j}^k(\vartheta(t)), \xi_{i,j}^k(\vartheta(t))]^T$, which is the prediction of $[p_{i,j}^k(t), v_{i,j}^k(t), \xi_{i,j}^k(t)]^T$ in the future. In addition, since the function $g_i(\cdot), i = 1, 2, 3$ contains the time delay $\tau(t)$, the previous analysis results of ESO in [48] cannot be directly applied to the PESOs in Equations (9) and (10), and the PESO-TVFTC is analyzed in the remainder of this paper.*

Up to now, the control law shown in Equation (8) has been able to be designed to make the system stable with the prediction of states and synthetic uncertainties. For multiple aircraft systems (7), the PESO-TVFTC protocol is proposed as follows :

$$
\begin{aligned}
u_{i,j}^k(t) &= c_{i,j}(\vartheta(t))\sigma_{i,j}({\varsigma_{i,j}^k}^T(t)\boldsymbol{P}\varsigma_{i,j}^k(t))\boldsymbol{K}\varsigma_{i,j}^k(t) \\
&+ \dot{h}_{vi,j}(\vartheta(t)) + \hat{\xi}_{N,j}^k(t) - \hat{\xi}_{i,j}^k(t)
\end{aligned}
\tag{11}
$$

where

$$
dc_{i,j}(\vartheta(t))/d\vartheta(t) = {\varsigma_{i,j}^k}^T(t)\boldsymbol{\Xi}\varsigma_{i,j}^k(t),
$$

$$
\varsigma_{i,j}^k(t) \triangleq \sum_{n=1}^{N-1} w_{in}[(\hat{\boldsymbol{s}}_{i,j}^k(t) - \boldsymbol{h}_{i,j}(\vartheta(t))) - (\hat{\boldsymbol{s}}_{n,j}^k(t) - \boldsymbol{h}_{n,j}(\vartheta(t)))]
$$

$$
+ w_{iN}(\hat{\boldsymbol{s}}_{i,j}^k(t) - \boldsymbol{h}_{i,j}(\vartheta(t)) - \hat{\boldsymbol{s}}_{N,j}^k(t))
$$

and $i \in \bar{F}, j \in \{x, y, z\}, k = i$. $u_{i,j}^k(t)$ are the control inputs of the outer-loop of the $i$th follower in the $j$ direction of its own flight path coordinate system at time $t$. $\sigma_{i,j}(\cdot)$ is a monotonic increasing function satisfying $\sigma_{i,j}(\omega) \geq 1$ with $\omega > 0$, $c_{i,j}(t)$ being the time-varying couple weights, $c_{i,j}(0) \geq 1$, $\hat{\boldsymbol{s}}_{i,j}^k(t), \hat{\xi}_{i,j}^k(t), \hat{\xi}_{N,j}^k(t)$ the estimation of states, synthetic uncertainties of followers, and synthetic uncertainties of the leader at time $\vartheta(t)$, respectively, and $w_{in}$ and $w_{iN}$ the weight of the directed topology between the $i$th follower and $n$th follower, and between the $i$th follower and the leader, respectively. $\boldsymbol{P}(\bar{\mu})$ is a unique positive definite matrix, $\boldsymbol{K} \in \mathbb{R}^{1 \times 2}$ is the low gain feedback matrix, and $\boldsymbol{\Xi} \in \mathbb{R}^{2 \times 2}$ is a gain matrix. It should be pointed out that since the input saturation constraints are embedded in systems (7), the peaking in the transient period of the PESO due to high gains can be avoided.

**Remark 7.** *The protocol (11) is a type of output feedback control law, which only requires the position information of the aircraft, and the velocities and the synthetic uncertainties can be estimated by the PESOs. In addition, compared with the consensus methods in [11–14], the protocol (11) is designed in a fully distributed fashion, which means that it uses only the position of its own and the neighboring aircraft without any requirements of global information such as the minimum eigenvalue of the Laplacian matrix.*

**Remark 8.** *Different from the protocol in [19], $\hat{\boldsymbol{s}}_{i,j}^k(t)$ and $\hat{\boldsymbol{s}}_{n,j}^k(t)$ in the protocol (11) are the estimations of $\boldsymbol{s}_{i,j}^k(\vartheta(t))$ and $\boldsymbol{s}_{n,j}^k(\vartheta(t))$, rather than $\boldsymbol{s}_{i,j}^k(t)$ and $\boldsymbol{s}_{n,j}^k(t)$, which means that control inputs can respond in advance by using the estimation of states at time $\vartheta(t)$, and thus, the problem of input time delay is addressed.*

*4.2. Low Gain Feedback Design Algorithm for Formation Tracking Control*

To keep the control inputs for the outer loop within the desired saturation constraints, the following algorithm is proposed based on low gain feedback to determine the parameters of protocol (11):

**Remark 9.** *Different from the design of the formation tracking control protocols in the existing literature, such as in [19], the low gain feedback method is embedded in Algorithm 1 to address the input saturation constraints problem. From Equations (11)–(14), one can select an appropriate $\bar{\mu}$ to make $\boldsymbol{K}$ and $\Xi$ small, so that the control inputs are within saturation constraints, namely $\left|u_{i,j}^k(t)\right| < \bar{M}_j$. The steps to obtain matrices $\boldsymbol{K}$ and $\Xi$ through Algorithm 1 are shown in Figure 3.*

---

**Algorithm 1** The parameters of protocol (11) can be specified in 4 steps:

---

**Step 1.** For systems (7) satisfying ANCBC, there exist a tuning parameter $\bar{\mu} \in (0,1]$ and a unique positive definite matrix $\boldsymbol{P}(\bar{\mu})$ satisfying the following algebraic Riccati equation:

$$\boldsymbol{P}(\bar{\mu})\boldsymbol{A} + \boldsymbol{A}^T\boldsymbol{P}(\bar{\mu}) - 2\boldsymbol{P}(\bar{\mu})\boldsymbol{B}\boldsymbol{B}^T\boldsymbol{P}(\bar{\mu}) + \bar{\mu}\boldsymbol{I} = 0. \tag{12}$$

The solution to Equation (12), $\boldsymbol{P}(\bar{\mu})$, is parameterized in $\bar{\mu}$, and $\boldsymbol{P}(\bar{\mu}) \to 0$ as $\bar{\mu} \to 0$.
**Step 2.** The low gain feedback matrix $\boldsymbol{K} \in \mathbb{R}^{1\times 2}$ can be specified by:

$$\boldsymbol{K} = -\boldsymbol{B}^T\boldsymbol{P}(\bar{\mu}). \tag{13}$$

**Step 3.** The gain matrix $\Xi \in \mathbb{R}^{2\times 2}$ can be specified by:

$$\Xi = \boldsymbol{P}(\bar{\mu})\boldsymbol{B}\boldsymbol{B}^T\boldsymbol{P}(\bar{\mu}). \tag{14}$$

**Step 4.** The monotonically increasing function $\sigma_{i,j}(\omega)$ can be designed as

$$\sigma_{i,j}(\omega) = (1+\omega)^\Lambda \tag{15}$$

where $\Lambda$ is a positive constant.

---

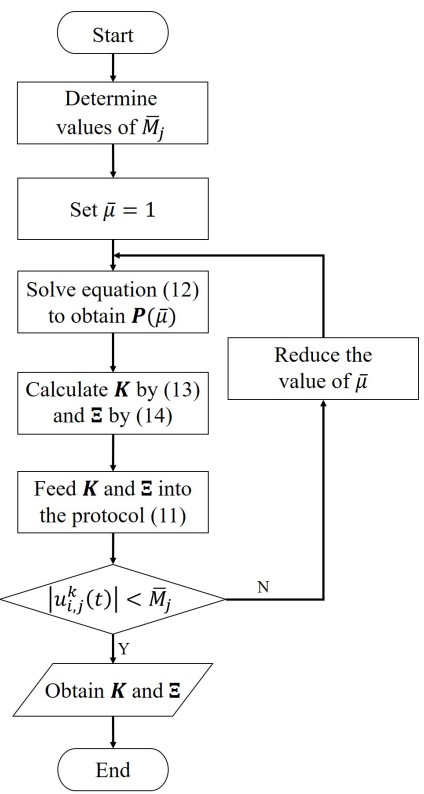

**Figure 3.** Steps to obtain matrices $\boldsymbol{K}$ and $\Xi$.

Up to this point, it is worthwhile to mention that by embedding the predictive values of states and synthetic uncertainties into the protocol (11), and by adjusting the parameter

$\bar{\mu}$ simultaneously, problems of input delay and input saturation constraint can be handled in conjunction.

*4.3. Stability Analysis*

Define the scaled estimation errors of states in PESOs (9) and (10) with:

$$
\begin{cases}
\eta_{pi,j}^k(t) = [p_{i,j}^k(\vartheta(t)) - \hat{p}_{i,j}^k(t)]/\varepsilon^2 \\
\eta_{vi,j}^k(t) = [v_{i,j}^k(\vartheta(t)) - \hat{v}_{i,j}^k(t)]/\varepsilon \\
\eta_{\xi i,j}^k(t) = [\xi_{i,j}^k(\vartheta(t)) - \hat{\xi}_{i,j}^k(t)]
\end{cases}
\tag{16}
$$

where $i, k \in \{1, 2, \ldots, N\}, j \in \{x, y, z\}$. When control inputs are within saturation constraints, from Equations (9), (10), and (16), the dynamics of these estimation errors satisfy:

$$
\begin{cases}
\varepsilon \dot{\eta}_{pi,j}^k(t) = \dot{\vartheta}(t)[\eta_{vi,j}^k(t) - g_1(\eta_{pi,j}^k(\varpi(t)))] \\
\varepsilon \dot{\eta}_{vi,j}^k(t) = \dot{\vartheta}(t)[\eta_{\xi i,j}^k(t) - g_2(\eta_{pi,j}^k(\varpi(t)))] \\
\varepsilon \dot{\eta}_{\xi i,j}^k(t) = \dot{\vartheta}(t)[\varepsilon \varphi(\vartheta(t)) - g_3(\eta_{pi,j}^k(\varpi(t)))]
\end{cases}
\tag{17}
$$

where

$$
\varphi(\varpi(t)) = \mathrm{d}(f(\vartheta(t), s_{i,j}^k(\vartheta(t)), \bar{z}_{i,j}^k(\vartheta(t)), \bar{d}_{i,j}^k(\vartheta(t))))/\mathrm{d}\vartheta(t)
$$

and $i, k \in \{1, 2, \ldots, N\}, j \in \{x, y, z\}$. Then the estimation errors of the PESO of the $i$th aircraft in the $j$ direction can be defined as $\boldsymbol{\eta}_{i,j}^k(t) = [\eta_{pi,j}^k(t), \eta_{vi,j}^k(t), \eta_{\xi i,j}^k(t)]^T \in \mathbb{R}^3$, where $i \in \{1, 2, \ldots, N\}, k = i$ and $j \in \{x, y, z\}$, and the total estimation errors of the multiple aircraft systems in the $j$ direction can be expressed as $\boldsymbol{\eta}_j(t) = [\boldsymbol{\eta}_{1,j}{}^T(t), \boldsymbol{\eta}_{2,j}{}^T(t), \ldots, \boldsymbol{\eta}_{N,j}{}^T(t)] \in \mathbb{R}^{3N(N-1)}$, where $i \in \{1, 2, \ldots, N\}, k \in \bar{F}, j \in \{x, y, z\}$, and $\boldsymbol{\eta}_{i,j}(t) = [\boldsymbol{\eta}_{i,j}^1{}^T(t), \boldsymbol{\eta}_{i,j}^2{}^T(t), \ldots, \boldsymbol{\eta}_{i,j}^{N-1}{}^T(t)]^T \in \mathbb{R}^{3(N-1)}$.

According to Definition 3 and protocol (11), as $\boldsymbol{\eta}_j(t)$ goes to zero, $\varsigma_{i,j}^k(t)$ can be the formation tracking errors of $i$th aircraft in the $j$ direction of its own flight path coordinate system, where $i \in \bar{F}, j \in \{x, y, z\}, k = i$. Then, the formation tracking errors of the multiple aircraft systems (7) can be denoted as $\varsigma_j(t) = [\varsigma_{1,j}^1{}^T(t), \varsigma_{2,j}^2{}^T(t), \ldots \varsigma_{N-1,j}^{N-1}{}^T(t)]^T \in \mathbb{R}^{2(N-1)}$, and then

$$
\varsigma_j(t) = (\boldsymbol{L}_1 \otimes \boldsymbol{I}_2)
\begin{bmatrix}
\hat{s}_{1,j}^1(t) - h_{1,j}(\vartheta(t)) - \hat{s}_{N,j}^1(t) \\
\hat{s}_{2,j}^2(t) - h_{2,j}(\vartheta(t)) - \hat{s}_{N,j}^2(t) \\
\vdots \\
\hat{s}_{N-1,j}^{N-1}(t) - h_{N-1,j}(\vartheta(t)) - \hat{s}_{N,j}^{N-1}(t)
\end{bmatrix}.
\tag{18}
$$

With protocol (11) and Algorithm 1, the PESOs (9) and (10) are designed such that the following assumptions are satisfied:

**Assumption 8 ([45]).** *The functions $g_i(\cdot), i = 1, 2, 3$ are global Lipschitz with a Lipschitz constant $\bar{K}_1$ and initial conditions $g_i(0) = 0$. For all $\boldsymbol{\eta}_{i,j}^k(t) \in \mathbb{R}^3$, there exist continuous, positive definite, and radially unbounded functions $\bar{V}_1(\cdot), \bar{W}_1(\cdot) : \mathbb{R}^3 \to \mathbb{R}^+$ and positive constants $\bar{c}_{11}, \bar{c}_{12}, \bar{c}_{13}, \bar{c}_{14}$ and $\bar{N}_1$ such that:*

$$
\bar{c}_{11}\left\|\boldsymbol{\eta}_{i,j}^k(t)\right\|^2 \leq \bar{V}_1(\boldsymbol{\eta}_{i,j}^k(t)) \leq \bar{c}_{12}\left\|\boldsymbol{\eta}_{i,j}^k(t)\right\|^2,
$$

$$
\bar{c}_{13}\left\|\boldsymbol{\eta}_{i,j}^k(t)\right\|^2 \leq \bar{W}_1(\boldsymbol{\eta}_{i,j}^k(t)) \leq \bar{c}_{14}\left\|\boldsymbol{\eta}_{i,j}^k(t)\right\|^2,
$$

$$
(\eta_{vi,j}^k(t) - g_1(\eta_{pi,j}^k(t)))\frac{\partial \bar{V}_1(\boldsymbol{\eta}_{i,j}^k(t))}{\partial \eta_{pi,j}^k(t)}
$$
$$
+ (\eta_{\xi i,j}^k(t) - g_2(\eta_{pi,j}^k(t)))\frac{\partial \bar{V}_1(\boldsymbol{\eta}_{i,j}^k(t))}{\partial \eta_{vi,j}^k(t)}
$$
$$
- g_3(\eta_{pi,j}^k(t))\frac{\partial \bar{V}_1(\boldsymbol{\eta}_{i,j}^k(t))}{\partial \eta_{\xi i,j}^k(t)} \leq -\bar{W}_1(\boldsymbol{\eta}_{i,j}^k(t)),
$$

$$\max\left\{\left|\frac{\partial \bar{V}_1(\boldsymbol{\eta}_{i,j}^k(t))}{\partial \eta_{pi,j}^k(t)}\right|, \left|\frac{\partial \bar{V}_1(\boldsymbol{\eta}_{i,j}^k(t))}{\partial \eta_{vi,j}^k(t)}\right|, \left|\frac{\partial \bar{V}_1(\boldsymbol{\eta}_{i,j}^k(t))}{\partial \eta_{\varsigma i,j}^k(t)}\right|\right\}$$
$$\leq \bar{N}_1\left\|\boldsymbol{\eta}_{i,j}^k(t)\right\|.$$

**Remark 10.** *Assumption 8 can be found in [45,48]. This assumption is not restrictive and can be satisfied by properly designing the PESOs (9) and (10).*

Then, we use Theorem 1 to show the stability of the PESO, and use Theorem 2 to show the stability of multiple aircraft systems with the formation controller. Before that, three lemmas that are used during the proof of the theorems are given. Lemma 4 is used to show the control inputs are within input saturation constraints, Lemma 5 is used to prove the estimation errors of the PESO will converge to a small positive constant, and Lemma 6 is used to show that $\varsigma_{i,j}^k(t)$ in (11) is uniformly bounded.

**Lemma 4.** *For multiple aircraft systems (7) satisfying the ANCBC and PESO-TVFTC protocol (11), $\bar{\mu}^* \in (0,1]$ and a bounded initial state set $\boldsymbol{\Omega}_0$ exist such that if $s_{i,j}^k(0), \hat{s}_{i,j}^k(0) \in \boldsymbol{\Omega}_0$, $i \in \{1,2,\ldots,N\}, k \in \bar{F}, j \in \{x,y,z\}$, then for any $\bar{\mu} \in (0,\bar{\mu}^*]$,*

$$\sup_{t\in[0,\infty),i\in\{1,2,\ldots,N\},k\in\bar{F},j\in\{x,y,z\}}\left|u_{i,j}^k(t)\right| < \bar{M}_j. \tag{19}$$

**Proof.** Let $\bar{V}_{\bar{\mu}}(\boldsymbol{\varsigma}_j(t), \boldsymbol{\eta}_j(t)) : \mathbb{R}^{2(N-1)} \times \mathbb{R}^{3N(N-1)} \to \mathbb{R}^+$ be the Lyapunov function of the system consisting of (7), (9), and (10). Since $\boldsymbol{\Omega}_0$ is bounded and $P(\bar{\mu}) \to 0$ and $c_{i,j}(\vartheta(t)) \to c_{i,j}(\vartheta(0))$ as $\bar{\mu} \to 0$, $\kappa_1 > 0$ exists, satisfying

$$\kappa_1 \geq \sup_{\bar{\mu}\in(0,1],s_{i,j}^k(0),\hat{s}_{i,j}^k(0)\in\boldsymbol{\Omega}_0} \bar{V}_{\bar{\mu}}(\boldsymbol{\varsigma}_j(0), \boldsymbol{\eta}_j(0)) \tag{20}$$

where $i \in \{1,2,\ldots,N\}, k \in \bar{F}, j \in \{x,y,z\}$. Let

$$\begin{aligned}\bar{W}_{\bar{\mu}}(\kappa_1) &\overset{\Delta}{=} \{\boldsymbol{\varsigma}_j(t) \in \mathbb{R}^{2(N-1)}, \boldsymbol{\eta}_j(t) \in \mathbb{R}^{3N(N-1)}| \\ &\quad \bar{V}_{\bar{\mu}}(\boldsymbol{\varsigma}_j(t), \boldsymbol{\eta}_j(t)) \leq \kappa_1, i \in \{1,2,\ldots,N\}, \\ &\quad k \in \bar{F}, j \in \{x,y,z\}\}. \end{aligned} \tag{21}$$

For any $t \in [0,\infty)$, according to the protocol (11), Algorithm 1, and Assumptions 2–7, when $\boldsymbol{\varsigma}_j(t), \boldsymbol{\eta}_j(t) \in \bar{W}_{\bar{\mu}}(\kappa_1)$, since $\lim_{\bar{\mu}\to 0} P(\bar{\mu}) = 0$, $\lim_{\bar{\mu}\to 0} c_{i,j}(\vartheta(t)) = c_{i,j}(\vartheta(0))$, and system (7) is ANCBC, $\bar{\mu}^* \in (0,1]$ exists such that for any $\bar{\mu} \in (0,\bar{\mu}^*]$, Equation (19) holds. $\square$

**Lemma 5.** *Suppose Assumptions 8 and Equation (19) hold. If $\varepsilon$-independent positive constants $\bar{N}_2$ and $t^*$ exist such that for any $t \in [0,t^*]$, $|\varphi(\vartheta(t))| \leq \bar{N}_2$ holds, then an $\varepsilon$-independent positive constant $\kappa$ exists such that for any $\varepsilon \geq \kappa \bar{h}$ and $t \in [0,t^*]$,*

$$\left\|\bar{\boldsymbol{\eta}}_{ti,j}^k\right\|_{\sup} \leq \sqrt{\frac{\bar{c}_{22}}{\bar{c}_{21}}}\left\|\bar{\boldsymbol{\eta}}_{0i,j}^k\right\|_{\sup} e^{-\frac{\bar{c}_{23}}{2\bar{c}_{22}\varepsilon}t} + \frac{2\bar{c}_{22}\bar{c}_{24}\vartheta_2\bar{N}_2}{\bar{c}_{21}\bar{c}_{23}}\varepsilon \tag{22}$$

*where $\bar{c}_{21}, \bar{c}_{22}, \bar{c}_{23}$ and $\bar{c}_{24}$ are positive constants, $\bar{\boldsymbol{\eta}}_{ti,j}^k = \boldsymbol{\eta}_{i,j}^k(t+\omega), \omega \in [-\bar{h},0], i \in \{1,2,\ldots,N\}, k \in \bar{F}, j \in \{x,y,z\}$, and $\left\|\bar{\boldsymbol{\eta}}_{ti,j}^k\right\|_{\sup} = \sup_{\omega\in[-\bar{h},0]}\left\|\boldsymbol{\eta}_{i,j}^k(t+\omega)\right\|$.*

**Proof.** See [45]. $\square$

Before giving the next Lemma, two compact sets are defined:

$$\Pi_0 = \left\{ \varsigma_j(t) \in \mathbb{R}^{2(N-1)} : \bar{V}_2(\varsigma_j(t)) \leq \zeta_0 \right\},$$
$$\Pi_1 = \left\{ \varsigma_j(t) \in \mathbb{R}^{2(N-1)} : \bar{V}_2(\varsigma_j(t)) \leq \zeta_1 \right\}$$

where $j \in \{x, y, z\}$, $\bar{V}_2(\cdot) \in \mathbb{R}^{2(N-1)} \to \mathbb{R}^+$ is a continuous, positive definite, and radially unbounded function, and $\zeta_0, \zeta_1$ are positive constants satisfying

$$\zeta_0 = \sup_{\|\varsigma_j(t)\| \leq \|\varsigma_j(0)\|} \bar{V}_2(\varsigma_j(t)) + 1$$

and $\zeta_1 \in (\zeta_0, \infty)$.

The feasibility conditions of time-varying formation tracking of multiple aircraft systems are given as follows:

$$(h_{vi,j}(t) - \dot{h}_{pi,j}(t)) = 0, i \in \bar{F}, j \in \{x, y, z\}. \tag{23}$$

**Lemma 6.** *For multiple aircraft systems* (7)*, considering PESOs* (9) *and* (10) *and the protocol* (11) *designed by Algorithm* 1*, if Assumptions* 4–8 *hold and Equation* (19) *and formation tracking feasibility conditions* (23) *are satisfied, then for any* $\varsigma_j(0) \in \Pi_0$*, there exist a small positive constant* $\varepsilon_1^*$ *and an $\varepsilon$-independent positive constant* $\kappa$ *, such that if* $\bar{h} < \varepsilon_1^*/\kappa$*,* $\varsigma_j(t) \in \Pi_1$ *for any* $\varepsilon \in (\kappa\bar{h}, \varepsilon_1^*)$ *and* $t > 0$.

**Proof.** See Appendix A. □

From Lemma 6, the Lyapunov function of the formation tracking errors is bounded. Up to this point, the main results of this paper can be given as follows:

**Theorem 1.** *(Stability of the PESO) For multiple aircraft systems* (7) *with input time delay and saturation and the PESOs* (9) *and* (10)*, if Assumptions* 1–8 *hold and the initial conditions* $s_{i,j}^k(0), \hat{s}_{i,j}^k(0) \in \Omega_0$ *are satisfied, then for any* $\varsigma_j(\vartheta(0)) \in \Pi_0$*, there exist* $\bar{\mu}^* \in (0, 1]$*, a small positive constant* $\varepsilon_1^*$*, and an $\varepsilon$-independent positive constant* $\kappa$ *such that if* $\bar{h} < \varepsilon_1^*/\kappa$*, for any* $\bar{\mu} \in (0, \bar{\mu}^*]$*,* $\varepsilon \in (\kappa\bar{h}, \varepsilon_1^*)$*, and* $t \in [0, \infty)$*,*

$$\begin{cases} \sup_{t \in [T(\varepsilon), \infty)} \left| p_{i,j}^k(\vartheta(t)) - \hat{p}_{i,j}^k(t) \right| = O(\varepsilon^3) \\ \sup_{t \in [T(\varepsilon), \infty)} \left| v_{i,j}^k(\vartheta(t)) - \hat{v}_{i,j}^k(t) \right| = O(\varepsilon^2) \\ \sup_{t \in [T(\varepsilon), \infty)} \left| \xi_{i,j}^k(\vartheta(t)) - \hat{\xi}_{i,j}^k(t) \right| = O(\varepsilon) \end{cases} \tag{24}$$

*where* $i \in \{1, 2, \ldots, N\}$*,* $k \in \bar{F}$*,* $j \in \{x, y, z\}$*,* $\Omega_0$ *is a bounded initial state set, and* $T(\varepsilon) \to 0$ *holds if* $\varepsilon \to 0$*.*

**Proof.** Firstly, from Lemma 4, if the initial conditions $s_{i,j}^k(0), \hat{s}_{i,j}^k(0) \in \Omega_0$ are satisfied, one can select an appropriate $\bar{\mu}$ such that Equation (19) holds, which means that control inputs of the outer-loop are within input saturation constraints. Secondly, from Lemma 4, one can obtain that for any $\varepsilon \in (\kappa\bar{h}, \varepsilon_1^*)$ and $t \in [0, \infty)$, $\bar{V}_2(\varsigma_j(t)) \in \Pi_1$ holds. Thus, Equation (A4) holds for any $t \in [0, \infty)$. Lastly, from Equation (A4), by letting $T(\varepsilon) = -(2\bar{c}_{22}/\bar{c}_{23})\varepsilon \ln \varepsilon^3$ , Equation (24) holds. □

**Theorem 2.** *(Stability of the multiple aircraft systems) For multiple aircraft systems* (7) *with input time delay and saturation, consider PESOs* (9) *and* (10) *and protocol* (11) *designed by Algorithm* 1*.*

*If Assumptions 1–8 hold and the initial conditions $s_{i,j}^k(0), \hat{s}_{i,j}^k(0) \in \Omega_0$ are satisfied, then the feasibility conditions of time-varying formation tracking of multiple aircraft systems are*

$$(h_{vi,j}(t) - \dot{h}_{pi,j}(t)) = 0, i \in \bar{F}, j \in \{x, y, z\}$$

*and if the feasibility conditions hold, then for any $\bar{\mu} \in (0, \bar{\mu}^*]$, $\varepsilon \in (\kappa\bar{h}, \varepsilon_1^*)$ and $t > 0$, the multiple aircraft systems achieve time-varying formation tracking consensus.*

**Proof.** Firstly, from Lemma 4, if the initial conditions $s_{i,j}^k(0), \hat{s}_{i,j}^k(0) \in \Omega_0$ are satisfied, one can select an appropriate $\bar{\mu}$ such that Equation (19) holds, which means that control inputs of the outer-loop are within input saturation constraints. Secondly, from Lemma 6, one can obtain that for any $\varepsilon \in (\kappa\bar{h}, \varepsilon_1^*)$ and $t \in [0, \infty)$, $\varsigma_j(t)$ is bounded and $\bar{V}_2(\varsigma_j(t)) \in \Pi_1$ holds, which yields Equations (A6) and (A20). Thirdly, from Equations (A6) and (A20), one has

$$\lim_{t \to \infty} \left\| \varsigma_j(t) \right\| = O(\varepsilon). \tag{25}$$

Lastly, Equation (25), together with Equations (18) and (A6), shows that Equation (6) holds. According to Definition 3, the multiple aircraft systems achieve time-varying formation tracking consensus. $\square$

**Remark 11.** *From Equations (24) and (25), it can be obtained that the formation tracking performances of multiple aircraft systems (7) under PESO-TVFTC protocol (11) designed by Algorithm 1 depend on $\varepsilon$, and the smaller the $\varepsilon$ value, the better the formation tracking performances. In [48], theoretically, the value of $\varepsilon$ can be selected to be arbitrarily small for better performance. However, from Theorems 1 and 2 one can see that for multiple aircraft systems (7), the lower bound of $\varepsilon$ is restricted by time delay, which provides a guideline for the implementation of PESO-TVFTC protocols for systems with input time delay.*

**Remark 12.** *From the proof of Theorems 1 and 2, one can obtain that $\bar{V}_2(\varsigma_j(t)) \in \Pi_1$. Thus, it follows from Equation (A8) that $c_{i,j}(\vartheta(t)), i \in \bar{F}, j \in \{x, y, z\}$ are bounded. From $dc_{i,j}(\vartheta(t))/d\vartheta(t) > 0$, one can see that as $\varsigma_j(t)$ converges to $\varepsilon$, $c_{i,j}(\vartheta(t))$ will converge to some finite values. This characteristic is verified in the simulation.*

**Remark 13.** *As mentioned in Theorems 1 and 2, the multiple aircraft systems (7) are required to satisfy initial conditions $s_{i,j}^k(0), \hat{s}_{i,j}^k(0) \in \Omega_0$, which means that the formation tracking consensus in this paper is semi-global. Considering the fact that multiple aircraft systems have specific mission areas, which means that the initial positions and velocities of aircraft are bounded, the requirements for the initial conditions are reasonable.*

**Remark 14.** *As mentioned in Theorem 2, the time-varying formation commands are required to satisfy feasibility conditions (23). In fact, $h_{pi,j}(t)$ and $h_{vi,j}(t)$ represent formation commands for the positions and velocities of an aircraft, respectively, which means that $h_{vi,j}(t)$ and the derivative of $h_{pi,j}(t)$ have the same physical meaning. Thus, feasibility conditions (23) hold naturally.*

## 5. Simulation

In this section, the theoretical results proposed in this paper are applied to formation tracking flight and the flight through the valley slit of multiple fixed-wing aircraft formation, which demonstrates the effectiveness and innovations of the theoretical results.

In order to show the cumulative effect of input time delay in the interactive communication, the 0-1 weighted directed interaction topology $G$ for the multiple fixed-wing aircraft systems can be designed as Figure 4.

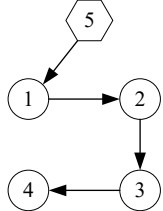

**Figure 4.** Directed interaction topology $G$.

The No.5 aircraft is the leader, which is a manned aircraft or unmanned aircraft tracking a target point, and the No.1 to No.4 aircraft are followers, which are unmanned and on autopilot.

The fixed-wing aircraft adopted in this paper are F-18 carrier aircraft with position and rate limits for the actuators and throttle, whose detailed configuration parameters can be found in [49]. The linear model of the F-18 aircraft is shown as follows:

$$A_{lon} = \begin{bmatrix} 0 & -38.6898 & -32.1700 & 0.0345 \\ 0 & -0.8562 & 0 & 0.9960 \\ 0 & 0 & 0 & 1.0000 \\ 0 & -2.1626 & 0 & -0.3103 \end{bmatrix}$$

$$B_{lon} = \begin{bmatrix} 4.8634 & 2.7387 \\ -0.1165 & -0.0005 \\ 0 & 0 \\ -8.8696 & 0 \end{bmatrix}$$

$$A_{lat} = \begin{bmatrix} -0.1576 & 0.0481 & -0.9982 & 0 & 0 & 0 & 0.0345 \\ -17.5966 & -2.0865 & 0.4986 & 0 & 0 & 0 & 0 \\ 3.3453 & -0.0187 & -0.1153 & 0 & 0 & 0 & 0 \\ 0 & 1.0000 & 0.0481 & 0 & 0 & 0 & 0 \\ 0 & 0 & 1.0012 & 0 & 0 & 0 & 0 \\ -0.1576 & 0 & 0.0006 & 0 & 0 & 0 & 0.0756 \\ 0 & 0.9988 & 0.0481 & 0 & 0 & 0 & 0 \end{bmatrix}$$

$$B_{lat} = \begin{bmatrix} -0.0043 & 0.0317 \\ 17.6941 & 2.4841 \\ -0.1946 & -2.1643 \\ 0 & 0 \\ 0 & 0 \\ -0.0043 & 0.0317 \\ 0 & 0 \end{bmatrix}$$

and $C_{L_0} = 0.2263, C_{D_0} = 0.0340, C_{L_\alpha} = 5.7289, C_{D_\alpha} \approx 0, T_0 = 23{,}785$ N, $\alpha_0 = 0.0481$ rad, $T_{\delta_T} = 44{,}450$ N/rad, $\rho = 0.4594$ kg/m$^3$, $S = 37.16$ m$^2$, $m = 16{,}220$ kg, and $g = 9.81$ m/s$^2$.

The delay time of the engine thrust is set as 0.1 s, which means that the input time delays in $u_{i,x}^k, u_{N,x}^k, i, k \in \bar{F}$ are $\tau = 0.1$ s. Considering the attitude tracking quickness and the trajectory tracking smoothness, the input saturation constraints of the outer-loop subsystem are $\bar{M}_x = 6.10$, $\bar{M}_y = 3.05$, and $\bar{M}_z = 1.52$.

In the outer-loop subsystem, the PESO-TVFTC protocol (11) designed by Algorithm 1 is adopted to realize time-varying formation tracking consensus. The PESO is designed with reference to [45] with $l_1 = 3, l_2 = 3, l_3 = 0.5$, and $\varepsilon = 0.3$. For the parameters in the protocol (11), $\Lambda$ in $x_k, y_k$ and $z_k$ directions are 0.0001, 0.00001, and 0.001, respectively. In the inner-loop subsystem, the incremental backstepping method [50] is adopted to realize stable attitude tracking.

The initial conditions are designed as follows: in the $x_k$ direction, $s_{i,x}^i(0) = [0, 283.98 \text{ m/s}]^T$ and $i = 1, 2, 3, 4, 5$; in the $y_k$ direction, $s_{1,y}^1(0) = [-46.10 \text{ m}, 0]^T$, $s_{2,y}^2(0) = [46.10 \text{ m}, 0]^T$, $s_{3,y}^3(0) = [-91.50 \text{ m}, 0]^T$, $s_{4,y}^4(0) = [91.50 \text{ m}, 0]^T$, and $s_{5,y}^5(0) = [0, 0]^T$; and in the $z_k$ direction, $s_{i,z}^i(0) = [0, 0]^T, i = 1, 2, 3, 4, 5$. The disturbances force in the $x_k$ direction is 1044.80 N.

The tracking target flies along the $x_k$ direction of the leader at a speed of 290.17 m/s. For $t \in [0, 10]$, the formation is assembling; for $t \in (10, 120]$, aircraft are forming a triangle formation; for $t \in (120, 200]$, the triangle formation is changing to a column formation; and for $t \in (200, 300]$, the formation is flying through a valley slit. In addition, the proposed

method in the paper is mainly tested through the changing of velocity in the $x_k$ direction; thus, $h_{pi,z}$ can be set as $0$, $i = 1, 2, 3, 4$. The formation commands can be designed as follows:

$$h_{p1,x}(t) = \begin{cases} -24.53, t \in [0, 120] \\ -0.31t + 12.67, t \in (120, 200] \\ -49.33, t \in (200, 300] \end{cases}$$

$$h_{p2,x}(t) = \begin{cases} -30.62, t \in [0, 120] \\ -0.88t + 74.98, t \in (120, 200] \\ -101.02, t \in (200, 300] \end{cases}$$

$$h_{p3,x}(t) = \begin{cases} -24.53, t \in [0, 120] \\ -1.56t + 162.67, t \in (120, 200] \\ -149.33, t \in (200, 300] \end{cases}$$

$$h_{p4,x}(t) = \begin{cases} -30.00, t \in [0, 120] \\ -2.13t + 225.60, t \in (120, 200] \\ -200.40, t \in (200, 300] \end{cases}$$

$$h_{p1,y}(t) = \begin{cases} 1.57t - 46.10, t \in [0, 10] \\ -30.40, t \in (10, 120] \\ 0.38t - 76.00, t \in (120, 200] \\ 0, t \in (200, 300] \end{cases}$$

$$h_{p2,y}(t) = \begin{cases} -1.57t + 46.10, t \in [0, 10] \\ 30.40, t \in (10, 120] \\ -0.38t + 76.00, t \in (120, 200] \\ 0, t \in (200, 300] \end{cases}$$

$$h_{p3,y}(t) = \begin{cases} 3.15t - 91.50, t \in [0, 10] \\ -60.00, t \in (10, 120] \\ 0.75t - 150.00, t \in (120, 200] \\ 0, t \in (200, 300] \end{cases}$$

$$h_{p4,y}(t) = \begin{cases} -3.15t + 91.50, t \in [0, 10] \\ 60.00, t \in (10, 120] \\ -0.75t + 150.00, t \in (120, 200] \\ 0, t \in (200, 300] \end{cases}$$

$$h_{pi,z}(t) = 0, t \in [0, 300], i = 1, 2, 3, 4.$$

Since input time delays mainly appear in $u_{i,x}^k$, $u_{N,x}^k$, $i, k \in \bar{F}$, this paper mainly shows the simulation results in the $x_k$ direction.

From Figure 5, one can see that the formation change of the multiple aircraft systems from a triangle to a column formation has been completed, so that the multiple aircraft systems can fly through a valley slit. Moreover, according to simulation experiments, in the process of the formation's assembly and change, the total formation tracking errors of the multiple aircraft systems in both $y_k$ and $z_k$ directions converge to zero.

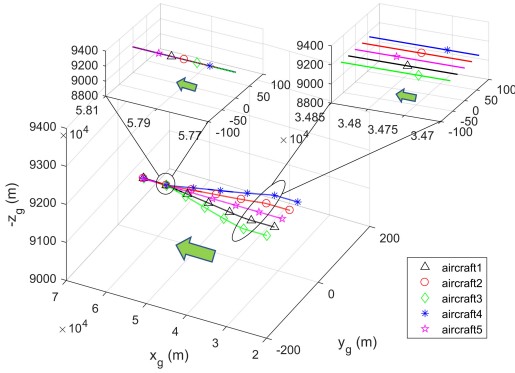

**Figure 5.** Formation change in flight at 100–220 s. (Subscript $g$ represents the earth coordinate system).

Figures 6 and 7 show the position formation errors and the velocity responses of the aircraft in the $x_k$ direction. Figures 6a and 7a show that the position formation errors and

the velocities oscillate when PESO-TVFTC is not used. The oscillations of the position formation errors and velocities are caused by input time delay. Specifically, due to the cumulative effect of delays in the directed interaction topology $G$, the oscillations of No.3 and No.4 aircraft that are in the third and fourth layers of the $G$, respectively, are more serious. However, when PESO-TVFTC is used, Figures 6b and 7b show that during the formation assembly and formation change, the position formation errors converge to the steady-state value at around $t = 75$ s and $t = 250$ s, respectively, and the velocities converge to the steady-state value at around $t = 55$ s and $t = 240$ s, respectively. In summary, due to the use of predictive information when using PESO-TVFTC, the negative effects of input time delays can be eliminated, and the accurate formation tracking of the multiple aircraft systems can be realized.

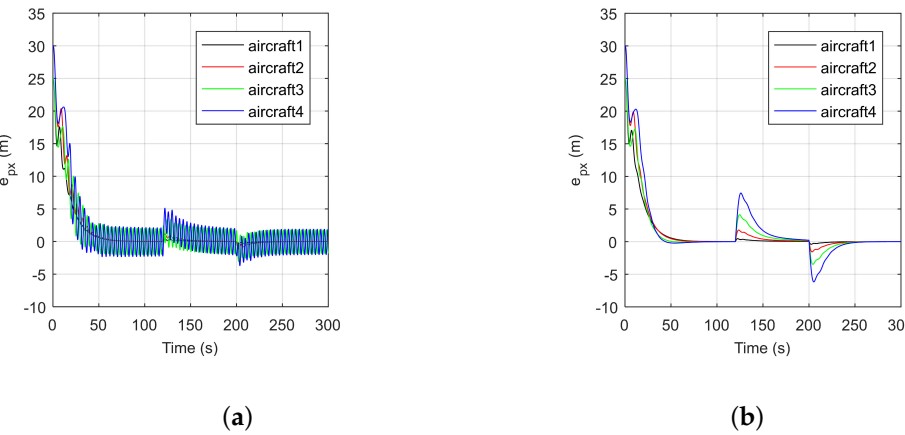

(**a**) (**b**)

**Figure 6.** Position formation tracking errors of all follows in the $x_k$ direction. (**a**) Formation tracking errors of ESO-TVFTC; (**b**) formation tracking errors of PESO-TVFTC.

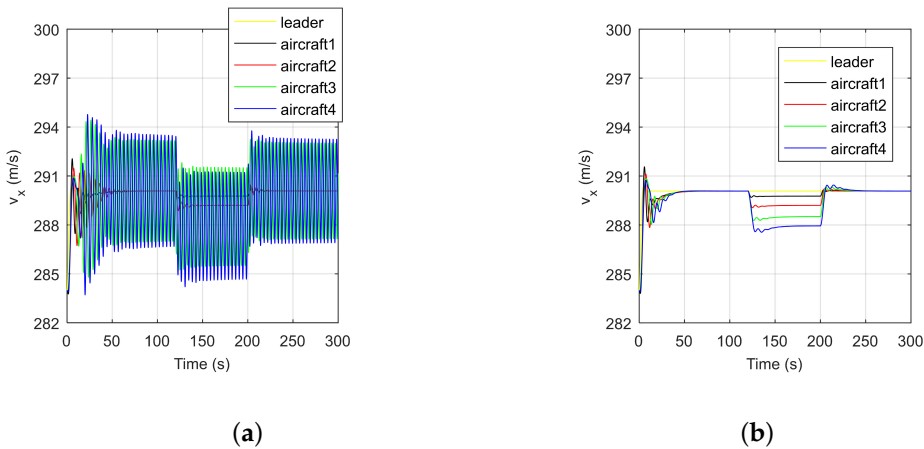

(**a**) (**b**)

**Figure 7.** Velocities of all aircraft in the in the $x_k$ direction. (**a**) Velocities of ESO-TVFTC; (**b**) velocities of PESO-TVFTC.

Figures 8 and 9 show the total formation tracking errors of the multiple aircraft systems and the estimation errors of PESO and ESO in the $x_k$ direction, respectively. From Figure 8, the root mean square of the total formation tracking errors within 280–300 s is 0.0018 using PESO-TVFTC and 3.2698 using ESO-TVFTC. Thus compared with ESO-TVFTC, the total number of formation tracking errors using PESO-TVFTC is significantly smaller and converges to a sufficiently small positive constant, which demonstrates the effectiveness of the proposed method. According to Figure 9, the estimation errors of the PESO are significantly smaller than those of the ESO, and converge to a sufficiently small positive constant. The results in Figures 8 and 9 verify Theorems 1 and 2.

Figure 10 shows that the coupling weights of all followers, which are controlled by the PESO-TVFTC protocol in the $x_k$ direction will converge to positive constants; that is, Remark 12 is verified.

Figure 11 illustrates the outer-loop control inputs of all followers controlled by the PESO-TVFTC protocol in the $x_k$ direction. It shows that the control inputs are all within the input saturation constraints. In addition, if the low gain feedback technology is not adopted, the control input given by Equation (11) may exceed the agility constraints of the aircraft, which can cause the aircraft to enter a stall or spin, resulting in the divergence of the multiple aircraft systems. Thus, the effectiveness of low gain feedback in Algorithm 1 is verified.

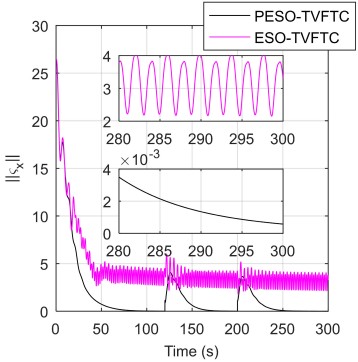

**Figure 8.** The total formation tracking errors of the multiple aircraft system in the $x_k$ direction.

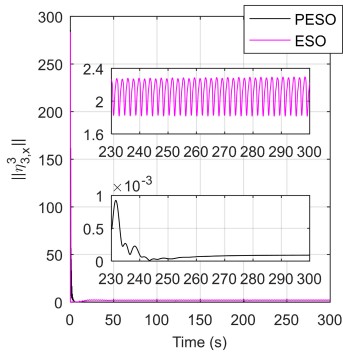

**Figure 9.** The estimation errors of PESO and ESO of the aircraft in the $x_k$ direction (taking No.3 aircraft for example).

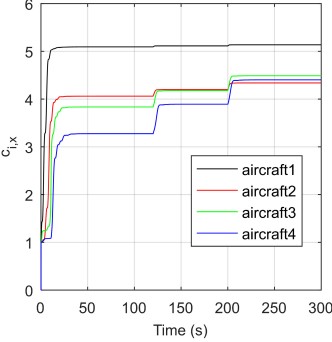

**Figure 10.** The coupling weights of all followers as in Equation (11) in the $x_k$ direction controlled by the PESO-TVFTC protocol.

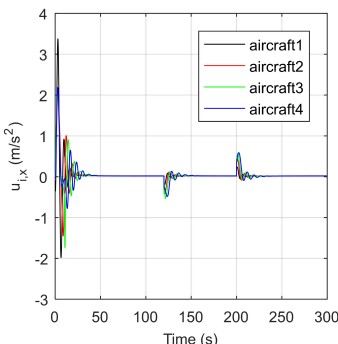

**Figure 11.** The outer-loop control inputs of all followers in the $x_k$ direction controlled by the PESO-TVFTC protocol, with $\bar{M}_x = 6.10$.

**Remark 15.** *It is worth mentioning that the PESO-TVFTC method proposed in this paper is also theoretically applicable to solving the formation tracking control problem subject to time-varying input delay given that the time delay information is known, e.g., time delay values or their derivatives are known. This conclusion can be deduced from Equations* (9) *and* (10).

## 6. Conclusions

In this paper, the predictive extended state observer-based fully distributed time-varying formation tracking control (PESO-TVFTC) protocol designed by using the low gain feedback technique is proposed to realize the fully distributed formation tracking consensus of multiple fixed-wing aircraft systems considering input time delay and saturation. It is shown that by using the proposed PESO-TVFTC protocol, the convergence of the formation tracking errors of multiple aircraft systems with uncertainties and input time delays can be guaranteed, and the lower bound of the PESO parameter $\varepsilon$ can be determined by the time delay. By applying the low gain feedback technique to design the proposed PESO-TVFTC protocol, the input saturation constraints in the outer-loop of the fixed-wing aircraft can be satisfied. Formation tracking flight scenarios are designed to simulate the formation assembly and formation change for multiple fixed-wing aircraft. According to simulation results, the PESO-TVFTC method can effectively address the engine time delay effects within the input saturation constraints when compared to the classical ESO-TVFTC method.

In future work, further improvement of the proposed method for solving the parameter drift problem caused by measurement noise is recommended. In addition, a potential field method and geometric method can be combined with the control law proposed in this paper to solve the obstacle avoidance problem of the whole formation system and the collision avoidance problem between aircraft.

**Author Contributions:** Methodology, L.S.; Writing—original draft, L.S.; Writing—review and editing, L.S., X.L., Y.D., J.J. and M.Z.; Supervision, W.T.; Funding acquisition, W.T. All authors have read and agreed to the published version of the manuscript.

**Funding:** This research was funded by [Aeronautical Science Foundation of China] grant number [20220048051001], [Aeronautical Science Foundation of China] grant number [20230013051002], and [Fundamental Research Funds for the Central Universities of China] grant number [YWF-23-SDHK-L-005].

**Data Availability Statement:** The raw data supporting the conclusions of this article will be made available by the authors on request.

**Conflicts of Interest:** The authors declare no conflict of interest.

## Appendix A. Proof of Lemma 6

**Proof.** Lemma 6 is proved by contradiction. Due to Equation (19), $\bar{u}_{i,j}^k(t) = u_{i,j}^k(t)$ is satisfied. Since $\varsigma_j(0)$ is an interior point of $\Pi_0$, control inputs $u_{i,j}^k(t)$ are bounded by $\bar{M}_j$, and Assumptions 4–7 hold, and there exists an $\varepsilon$-independent $t_0 > 0$ such that for any $t \in [0, t_0]$, $\varsigma_j(t) \in \Pi_0$ holds. Supposing that Lemma 6 is false, then $t_2 > t_1 > t_0$ exists such that

$$\begin{cases} \bar{V}_2(\varsigma_j(t_1)) = \zeta_0 \\ \bar{V}_2(\varsigma_j(t_2)) = \zeta_1 \\ \zeta_0 \le \bar{V}_2(\varsigma_j(t)) \le \zeta_1, t \in [t_1, t_2] \\ \bar{V}_2(\varsigma_j(t)) \le \zeta_1, t \in [0, t_2]. \end{cases} \tag{A1}$$

Moreover, the term $\varphi(\vartheta(t))$ in Equation (17) can be specified as:

$$\begin{aligned} \varphi(\vartheta(t)) =& \frac{\partial f(\cdot)}{\partial \vartheta(t)} + v_{i,j}^k(\vartheta(t)) \frac{\partial f(\cdot)}{\partial p_{i,j}^k(\vartheta(t))} \\ &+ (\xi_{i,j}^k(\vartheta(t)) + u_{i,j}^k(t)) \frac{\partial f(\cdot)}{\partial v_{i,j}^k(\vartheta(t))} \\ &+ f_0(\vartheta(t), s_{i,j}^k(\vartheta(t)), \bar{z}_{i,j}^k(\vartheta(t)), \bar{d}_{i,j}^k(\vartheta(t))) \frac{\partial f(\cdot)}{\partial \bar{z}_{i,j}^k(\vartheta(t))} \\ &+ \frac{d(\bar{d}_{i,j}^k(\vartheta(t)))}{d\vartheta(t)} \frac{\partial f(\cdot)}{\partial \bar{d}_{i,j}^k(\vartheta(t))}. \end{aligned} \tag{A2}$$

From Assumptions 5–7 and Equations (19) and (A1), one can obtain that for any $t \in [0, t_2]$, all terms in Equation (A2) are bounded. Thus, an $\varepsilon$-independent positive constant $\bar{N}_2$ exists such that

$$|\varphi(\vartheta(t))| \le \bar{N}_2, \forall t \in [0, t_2]. \tag{A3}$$

Next, we prove that a small positive constant $\varepsilon_1$ exists such that for any $\varepsilon \in (\kappa \bar{h}, \varepsilon_1)$, $\left\| \eta_{i,j}^k(t) \right\|$ converges to a sufficiently small positive constant in the time interval $[t_0, t_2]$. From Equation (A3) and Lemma 5, it can be obtained that for any $t \in [0, t_2]$,

$$\left\| \bar{\eta}_{ti,j}^k \right\|_{\sup} \le \sqrt{\frac{\bar{c}_{22}}{\bar{c}_{21}}} \left\| \bar{\eta}_{0i,j}^k \right\|_{\sup} e^{-\frac{\bar{c}_{23}}{2\bar{c}_{22}\varepsilon}t} + \frac{2\bar{c}_{22}\bar{c}_{24}\vartheta_2\bar{N}_2}{\bar{c}_{21}\bar{c}_{23}}\varepsilon. \tag{A4}$$

Based on Equation (A4), one can see that for any $t \in [t_0, t_2]$,

$$\left\| \bar{\eta}_{ti,j}^k \right\|_{\sup} \le \sqrt{\frac{\bar{c}_{22}}{\bar{c}_{21}}} \left\| \bar{\eta}_{0i,j}^k \right\|_{\sup} e^{-\frac{\bar{c}_{23}}{2\bar{c}_{22}\varepsilon}t_0} + \frac{2\bar{c}_{22}\bar{c}_{24}\vartheta_2\bar{N}_2}{\bar{c}_{21}\bar{c}_{23}}\varepsilon. \tag{A5}$$

From Equation (A5), one can obtain that a small positive constant $\varepsilon_1$ exists such that for any $\varepsilon \in [\kappa \bar{h}, \varepsilon_1]$ and $t \in [t_0, t_2]$,

$$\left\| \bar{\eta}_{ti,j}^k \right\|_{\sup} = O(\varepsilon) \tag{A6}$$

which means that $\left\| \eta_{i,j}^k(t) \right\|$ converges to a small positive constant $\varepsilon$ in the time interval $[t_0, t_2]$.

Next, Equation (A1) is proved to be false for any $t \in [t_1, t_2]$.

Let $\xi_j(t) = [\xi_{1,j}^1(t), \xi_{2,j}^2(t), \dots, \xi_{N-1,j}^{N-1}(t)]^T$, and $\hat{\xi}_j(t) = [\hat{\xi}_{1,j}^1(t), \hat{\xi}_{2,j}^2(t), \dots, \hat{\xi}_{N-1,j}^{N-1}(t)]^T$, $j \in \{x, y, z\}$. From Equations (7), (9)–(11) and (19), one can obtain that

$$\begin{aligned} \dot{\varsigma}_j(t) =& \dot{\vartheta}(t)(I_{N-1} \otimes A + L_1 \tilde{C}_j \tilde{\sigma}_j \otimes BK)\varsigma_j(t) + \\ & \dot{\vartheta}(t)(L_1 \otimes I_2)\Gamma_j + \dot{\vartheta}(t)(L_1 \otimes B)(\xi_j(\vartheta(t)) - \hat{\xi}_j(t)) + \\ & \dot{\vartheta}(t)(L_2 \otimes B)(\xi_{N,j}^i(\vartheta(t)) - \hat{\xi}_{N,j}^i(t)). \end{aligned} \tag{A7}$$

For simplicity and convenience, denote $\varsigma_j(t)$ as $\varsigma_j$, $\varsigma_{i,j}^k(t)$ as $\varsigma_{i,j}^k$, $c_{i,j}(\vartheta(t))$ as $c_{i,j}$, and $P(\bar{\mu})$ as $P$. In Equation (A7), $j \in \{x, y, z\}$ and

$$\tilde{C}_j \overset{\Delta}{=} diag\{c_{1,j}, c_{2,j}, \ldots, c_{N-1,j}\},$$

$$\tilde{\sigma}_j \overset{\Delta}{=} diag\{\sigma_{1,j}(\varsigma_{1,j}^{1\,T} P \varsigma_{1,j}^1), \sigma_{2,j}(\varsigma_{2,j}^{2\,T} P \varsigma_{2,j}^2), \ldots,$$
$$\sigma_{N-1,j}(\varsigma_{N-1,j}^{N-1\,T} P \varsigma_{N-1,j}^{N-1})\},$$

$$\Gamma_j \overset{\Delta}{=} [h_{v1,j}(\vartheta(t)), 0, h_{v2,j}(\vartheta(t)), 0, \ldots, h_{vN-1,j}(\vartheta(t)), 0]^{\mathrm{T}}$$
$$-[\dot{h}_{p1,j}(\vartheta(t)), 0, \dot{h}_{p2,j}(\vartheta(t)), 0, \ldots, \dot{h}_{pN-1,j}(\vartheta(t)), 0]^{\mathrm{T}}.$$

Consider the following Lyapunov function candidate:

$$\bar{V}_2(\varsigma_j) = \sum_{i=1}^{N-1} c_{i,j} r_i \int_0^{\varsigma_{i,j}^{i\mathrm{T}} P \varsigma_{i,j}^i} \sigma_{i,j}(\omega) d\omega + \frac{\bar{\lambda}_0}{\bar{m}} \sum_{i=1}^{N-1} \bar{c}_{i,j}^2 \tag{A8}$$

where $i \in \bar{F}, j \in \{x, y, z\}$, $\bar{c}_{i,j} = c_{i,j} - \bar{v}_1$, $\bar{m} > 3$, and $\bar{v}_1 > 0$, which will be determined later. $\bar{\lambda}_0$ is a positive constant. From the properties of $c_{i,j}(\cdot)$ and $\sigma_{i,j}(\cdot)$, one can see that $\bar{V}_2(\varsigma_j) > 0$ and is continuous and radially unbounded.

If the feasibility condition (23) is satisfied in the time interval $t \in [t_1, t_2]$, one can see $\Gamma_j = 0$. From the properties of $c_{i,j}(\cdot), \sigma_{i,j}(\cdot), R, L_1$ and the fact that $P$ and $BB^{\mathrm{T}}$ are not zero matrixes, one has

$$\varsigma_j^{\mathrm{T}}(\tilde{C}_j \tilde{\sigma}_j RL_1 \otimes P)\Gamma_j$$
$$\leq \tfrac{1}{2}\varsigma_j^{\mathrm{T}}(\tilde{C}_j \tilde{\sigma}_j (RL_1 + L_1^{\mathrm{T}} R)\tilde{C}_j \tilde{\sigma}_j \otimes PBB^{\mathrm{T}} P)\varsigma_j \tag{A9}$$
$$-\tfrac{1}{2}\varsigma_j^{\mathrm{T}}(\tilde{C}_j \tilde{\sigma}_j (\tfrac{3}{\bar{m}}\bar{\lambda}_0 I_{N-1})\tilde{C}_j \tilde{\sigma}_j \otimes PBB^{\mathrm{T}} P)\varsigma_j.$$

The derivative of $\bar{V}_2(\varsigma_j)$ with respect to $t$ can be obtained as follows:

$$\dot{\bar{V}}_2(\varsigma_j) = \sum_{i=1}^{N-1} c_{i,j} r_i \sigma_{i,j}(\varsigma_{i,j}^{i\mathrm{T}} P \varsigma_{i,j}^i) \varsigma_{i,j}^{i\mathrm{T}} P \dot{\varsigma}_{i,j}^i$$
$$+ \sum_{i=1}^{N-1} c_{i,j} r_i \sigma_{i,j}(\varsigma_{i,j}^{i\mathrm{T}} P \varsigma_{i,j}^i) \dot{\varsigma}_{i,j}^{i\mathrm{T}} P \varsigma_{i,j}^i$$
$$+ \sum_{i=1}^{N-1} \dot{\vartheta}(t)\frac{dc_{i,j}}{d\vartheta(t)} r_i \int_0^{\varsigma_{i,j}^{i\mathrm{T}} P \varsigma_{i,j}^i} \sigma_{i,j}(\omega) d\omega \tag{A10}$$
$$+ \frac{\bar{\lambda}_0}{\bar{m}} \sum_{i=1}^{N-1} \dot{\vartheta}(t) 2\bar{c}_{i,j} \varsigma_{i,j}^{i\mathrm{T}} PBB^{\mathrm{T}} P \varsigma_{i,j}^i.$$

From Equations (A7) and (A9), and Algorithm 1, it can be obtained that

$$\sum_{i=1}^{N-1} c_{i,j} r_i \sigma_{i,j}(\varsigma_{i,j}^{i\mathrm{T}} P \varsigma_{i,j}^i) \varsigma_{i,j}^{i\mathrm{T}} P \dot{\varsigma}_{i,j}^i$$
$$+ \sum_{i=1}^{N-1} c_{i,j} r_i \sigma_{i,j}(\varsigma_{i,j}^{i\mathrm{T}} P \varsigma_{i,j}^i) \dot{\varsigma}_{i,j}^{i\mathrm{T}} P \varsigma_{i,j}^i$$
$$= \varsigma_j^{\mathrm{T}}(\tilde{C}_j \tilde{\sigma}_j R \otimes P)\dot{\varsigma}_j + \dot{\varsigma}_j^{\mathrm{T}}(\tilde{C}_j \tilde{\sigma}_j R \otimes P)\varsigma_j \tag{A11}$$
$$\leq \dot{\vartheta}(t)\varsigma_j^{\mathrm{T}}(\tilde{C}_j \tilde{\sigma}_j R \otimes (A^{\mathrm{T}} P + PA)$$
$$- \tfrac{3}{\bar{m}}\bar{\lambda}_0 \tilde{C}_j^2 \tilde{\sigma}_j^2 \otimes PBB^{\mathrm{T}} P)\varsigma_j$$
$$+ 4\dot{\vartheta}(t)\varsigma_j^{\mathrm{T}}(\tilde{C}_j \tilde{\sigma}_j RL_1 \otimes PB)\mathbf{1}_{N-1}\left\|\bar{\eta}_{ti,j}^i\right\|_{\sup}$$

where $i \in \bar{F}, j \in \{x, y, z\}$. From the properties of $c_{i,j}(\cdot), \sigma_{i,j}(\cdot)$, and Lemma 3, one can obtain that

$$
\begin{aligned}
&\sum_{i=1}^{N-1} \dot{\vartheta}(t) \frac{dc_{i,j}}{d\vartheta(t)} r_i \int_0^{\varsigma_{i,j}^{iT} \boldsymbol{P} \varsigma_{i,j}^i} \sigma_{i,j}(\omega) d\omega \\
&\leq \sum_{i=1}^{N-1} \dot{\vartheta}(t) \frac{dc_{i,j}}{d\vartheta(t)} r_i \sigma_{i,j}(\varsigma_{i,j}^{iT} \boldsymbol{P} \varsigma_{i,j}^i) \varsigma_{i,j}^{iT} \boldsymbol{P} \varsigma_{i,j}^i \\
&\leq \sum_{i=1}^{N-1} \dot{\vartheta}(t) \frac{dc_{i,j}}{d\vartheta(t)} \left( \frac{r_i^3}{3\bar{\lambda}_0^2} + \frac{2}{3} \bar{\lambda}_0 \sigma_{i,j}^{1.5} (\varsigma_{i,j}^{iT} \boldsymbol{P} \varsigma_{i,j}^i) (\varsigma_{i,j}^{iT} \boldsymbol{P} \varsigma_{i,j}^i)^{1.5} \right) \\
&\leq \sum_{i=1}^{N-1} \dot{\vartheta}(t) \frac{dc_{i,j}}{d\vartheta(t)} \left( \frac{r_i^3}{3\bar{\lambda}_0^2} + \frac{2}{3} \bar{\lambda}_0 \sigma_{i,j}^{1.5} (\varsigma_{i,j}^{iT} \boldsymbol{P} \varsigma_{i,j}^i) (1 + \varsigma_{i,j}^{iT} \boldsymbol{P} \varsigma_{i,j}^i)^{1.5} \right) \\
&= \sum_{i=1}^{N-1} \dot{\vartheta}(t) \left( \frac{r_i^3}{3\bar{\lambda}_0^2} + \frac{2}{3} \bar{\lambda}_0 \sigma_{i,j}^{1.5 + \frac{1.5}{\Lambda}} (\varsigma_{i,j}^{iT} \boldsymbol{P} \varsigma_{i,j}^i) \right) \varsigma_{i,j}^{iT} \boldsymbol{P} \boldsymbol{B} \boldsymbol{B}^{\mathrm{T}} \boldsymbol{P} \varsigma_{i,j}^i.
\end{aligned}
\tag{A12}
$$

Substituting Equations (A11) and (A12) into Equation (A10) yields

$$
\begin{aligned}
\dot{V}_2(\varsigma_j) \leq{} & \dot{\vartheta}(t) \varsigma_j^{\mathrm{T}} (\tilde{\boldsymbol{C}}_j \tilde{\boldsymbol{\sigma}}_j \boldsymbol{R} \otimes (\boldsymbol{A}^{\mathrm{T}} \boldsymbol{P} + \boldsymbol{P} \boldsymbol{A})) \varsigma_j \\
&+ 4\dot{\vartheta}(t) \varsigma_j^{\mathrm{T}} (\tilde{\boldsymbol{C}}_j \tilde{\boldsymbol{\sigma}}_j \boldsymbol{R} \boldsymbol{L}_1 \otimes \boldsymbol{P} \boldsymbol{B}) \mathbf{1}_{N-1} \left\| \bar{\boldsymbol{\eta}}_{ti,j}^i \right\|_{\sup} \\
&- \dot{\vartheta}(t) \sum_{i=1}^{N-1} \left( \frac{3}{\bar{m}} \bar{\lambda}_0 c_{i,j}^2 \sigma_{i,j}^2 (\varsigma_{i,j}^{iT} \boldsymbol{P} \varsigma_{i,j}^i) - \frac{2}{\bar{m}} \bar{\lambda}_0 c_{i,j} \right) \\
&\times \varsigma_{i,j}^{iT} \boldsymbol{P} \boldsymbol{B} \boldsymbol{B}^{\mathrm{T}} \boldsymbol{P} \varsigma_{i,j}^i \\
&- \dot{\vartheta}(t) \sum_{i=1}^{N-1} \left( \frac{2}{\bar{m}} \bar{\lambda}_0 \bar{v}_1 - \frac{r_i^3}{3\bar{\lambda}_0^2} - \frac{2}{3} \bar{\lambda}_0 \sigma_{i,j}^{1.5 + \frac{1.5}{\Lambda}} (\varsigma_{i,j}^{iT} \boldsymbol{P} \varsigma_{i,j}^i) \right) \\
&\times \varsigma_{i,j}^{iT} \boldsymbol{P} \boldsymbol{B} \boldsymbol{B}^{\mathrm{T}} \boldsymbol{P} \varsigma_{i,j}^i.
\end{aligned}
\tag{A13}
$$

From Equation (A1), one can see that $\bar{V}_2(\varsigma_j)$ is bounded for any $t \in [t_1, t_2]$. Thus, $\bar{v}_1$ can be set as

$$
\bar{v}_1 \geq \frac{\bar{v}_2}{2} + \max_{t \in [t_1, t_2]} \left[ \frac{\bar{m} r_i^3}{6\bar{\lambda}_0^3} + \frac{\bar{m}}{3} \sigma_{i,j}^{1.5 + \frac{1.5}{\Lambda}} (\varsigma_{i,j}^{iT} \boldsymbol{P} \varsigma_{i,j}^i) \right]
\tag{A14}
$$

where $\bar{v}_2$ is a positive constant satisfying

$$
\sqrt{\bar{v}_2} \boldsymbol{I}_{N-1} \geq \frac{\bar{m}}{\bar{\lambda}_0} \boldsymbol{R} + \frac{\bar{m}}{2\bar{\lambda}_0} \boldsymbol{\Theta}
\tag{A15}
$$

and $\boldsymbol{\Theta}$ is a positive definite diagonal matrix and satisfies $\boldsymbol{\Theta} \geq 4 \boldsymbol{R} \boldsymbol{L}_1$. Then, taking the properties of $c_{i,j}(\cdot)$ and $\sigma_{i,j}(\cdot)$, by substituting Equations (A14) and (A15) into Equation (A13), it follows that

$$
\begin{aligned}
\dot{V}_2(\varsigma_j) \leq{} & \dot{\vartheta}(t) \varsigma_j^{\mathrm{T}} (\tilde{\boldsymbol{C}}_j \tilde{\boldsymbol{\sigma}}_j \boldsymbol{R} \otimes (\boldsymbol{A}^{\mathrm{T}} \boldsymbol{P} + \boldsymbol{P} \boldsymbol{A})) \varsigma_j \\
&- \dot{\vartheta}(t) \sum_{i=1}^{N-1} \left( \frac{1}{\bar{m}} \bar{\lambda}_0 c_{i,j}^2 \sigma_{i,j}^2 (\varsigma_{i,j}^{iT} \boldsymbol{P} \varsigma_{i,j}^i) + \frac{\bar{\lambda}_0}{\bar{m}} \bar{v}_2 \right) \varsigma_{i,j}^{iT} \boldsymbol{P} \boldsymbol{B} \boldsymbol{B}^{\mathrm{T}} \boldsymbol{P} \varsigma_{i,j}^i \\
&+ 4\dot{\vartheta}(t) \varsigma_j^{\mathrm{T}} (\tilde{\boldsymbol{C}}_j \tilde{\boldsymbol{\sigma}}_j \boldsymbol{R} \boldsymbol{L}_1 \otimes \boldsymbol{P} \boldsymbol{B}) \mathbf{1}_{N-1} \left\| \bar{\boldsymbol{\eta}}_{ti,j}^i \right\|_{\sup} \\
\leq{} & \dot{\vartheta}(t) \varsigma_j^{\mathrm{T}} (\tilde{\boldsymbol{C}}_j \tilde{\boldsymbol{\sigma}}_j \boldsymbol{R} \otimes (\boldsymbol{A}^{\mathrm{T}} \boldsymbol{P} + \boldsymbol{P} \boldsymbol{A})) \varsigma_j \\
&- \dot{\vartheta}(t) \varsigma_j^{\mathrm{T}} (2 \boldsymbol{R} \tilde{\boldsymbol{C}}_j \tilde{\boldsymbol{\sigma}}_j \otimes \boldsymbol{P} \boldsymbol{B} \boldsymbol{B}^{\mathrm{T}} \boldsymbol{P}) \varsigma_j \\
&- \dot{\vartheta}(t) \varsigma_j^{\mathrm{T}} (\boldsymbol{\Theta} \tilde{\boldsymbol{C}}_j \tilde{\boldsymbol{\sigma}}_j \otimes \boldsymbol{P} \boldsymbol{B} \boldsymbol{B}^{\mathrm{T}} \boldsymbol{P}) \varsigma_j \\
&+ 4\dot{\vartheta}(t) \varsigma_j^{\mathrm{T}} (\tilde{\boldsymbol{C}}_j \tilde{\boldsymbol{\sigma}}_j \boldsymbol{R} \boldsymbol{L}_1 \otimes \boldsymbol{P} \boldsymbol{B}) \mathbf{1}_{N-1} \left\| \bar{\boldsymbol{\eta}}_{ti,j}^i \right\|_{\sup}.
\end{aligned}
\tag{A16}
$$

Let $\dot{V}_2(\varsigma_j) \leq \dot{V}_{21}(\varsigma_j) + \dot{V}_{22}(\varsigma_j)$, where

$$
\begin{aligned}
\dot{V}_{21}(\varsigma_j) ={} & \dot{\vartheta}(t) \varsigma_j^{\mathrm{T}} (\tilde{\boldsymbol{C}}_j \tilde{\boldsymbol{\sigma}}_j \boldsymbol{R} \otimes (\boldsymbol{A}^{\mathrm{T}} \boldsymbol{P} + \boldsymbol{P} \boldsymbol{A})) \varsigma_j \\
&- \dot{\vartheta}(t) \varsigma_j^{\mathrm{T}} (2 \boldsymbol{R} \tilde{\boldsymbol{C}}_j \tilde{\boldsymbol{\sigma}}_j \otimes \boldsymbol{P} \boldsymbol{B} \boldsymbol{B}^{\mathrm{T}} \boldsymbol{P}) \varsigma_j,
\end{aligned}
\tag{A17}
$$

$$\dot{V}_{22}(\varsigma_j) = 4\dot{\vartheta}(t)\varsigma_j^{\mathrm{T}}(\tilde{C}_j\tilde{\sigma}_j RL_1 \otimes PB)\mathbf{1}_{N-1}\left\|\bar{\eta}_{ti,j}^i\right\|_{\sup} \\ -\dot{\vartheta}(t)\varsigma_j^{\mathrm{T}}(\Theta\tilde{C}_j\tilde{\sigma}_j \otimes PBB^{\mathrm{T}}P)\varsigma_j. \tag{A18}$$

Next, we prove $\dot{V}_2 < 0$. Firstly, $\dot{V}_{21}(\varsigma_j) < 0$ is proved as follows. Since

$$\dot{V}_{21}(\varsigma_j) = \dot{\vartheta}(t)\varsigma_j^{\mathrm{T}}(\tilde{C}_j\tilde{\sigma}_j R \otimes (A^{\mathrm{T}}P + PA - 2PBB^{\mathrm{T}}P))\varsigma_j \tag{A19}$$

one can obtain $\dot{V}_{21}(\varsigma_j) < 0$ by substituting Equation (12) into Equation (A19).

Secondly, $\dot{V}_{22}(\varsigma_j) \le 0$ is proved. Let $\bar{\lambda}_1 = \|PB\|_\infty$, and $\bar{\lambda}_2 > 0$ exists such that $\bar{\lambda}_2\left\|\varsigma_{i,j}^i\right\|^2 \le \varsigma_{i,j}^{i\mathrm{T}}PBB^{\mathrm{T}}P\varsigma_{i,j}^i$ for any $t \in [t_1, t_2]$. Then, one has

$$\begin{aligned} \dot{V}_{22}(\varsigma_j) &\le 4\dot{\vartheta}(t)\varsigma_j^{\mathrm{T}}(\tilde{C}_j\tilde{\sigma}_j RL_1 \otimes PB)\mathbf{1}_{N-1}\left\|\bar{\eta}_{ti,j}^i\right\|_{\sup} \\ &\quad -4\dot{\vartheta}(t)\varsigma_j^{\mathrm{T}}(\tilde{C}_j\tilde{\sigma}_j RL_1 \otimes PBB^{\mathrm{T}}P)\varsigma_j \\ &= 4\dot{\vartheta}(t)\sum_{i=1}^{N-1} c_{i,j}\sigma_{i,j}(\varsigma_{i,j}^{i\mathrm{T}}P\varsigma_{i,j}^i)r_i\bar{l}_{ii} \\ &\quad \times(\varsigma_{i,j}^{i\mathrm{T}}PB\left\|\bar{\eta}_{ti,j}^i\right\|_{\sup} - \varsigma_{i,j}^{i\mathrm{T}}PBB^{\mathrm{T}}P\varsigma_{i,j}^i) \\ &\le 4\dot{\vartheta}(t)\sum_{i=1}^{N-1} c_{i,j}\sigma_{i,j}(\varsigma_{i,j}^{i\mathrm{T}}P\varsigma_{i,j}^i)r_i\bar{l}_{ii} \\ &\quad \times(\bar{\lambda}_1\sqrt{2}\left\|\varsigma_{i,j}^i\right\|\left\|\bar{\eta}_{ti,j}^i\right\|_{\sup} - \bar{\lambda}_2\left\|\varsigma_{i,j}^i\right\|^2) \end{aligned} \tag{A20}$$

where $\bar{l}_{ij}$ denotes the element in the $i$th row and $j$th column of the matrix $L_1$. From Equations (A4) and (A20), one can see that in the time interval $[t_1, t_2]$, one can select $\varepsilon$ to be sufficiently small such that $\left\|\bar{\eta}_{ti,j}^i\right\|_{\sup} \le \left\|\varsigma_{i,j}^i\right\|\bar{\lambda}_2 / (\sqrt{2}\bar{\lambda}_1)$, $i \in \bar{F}$, namely $\dot{V}_{22}(\varsigma_j) \le 0$. Thus, a sufficiently small positive constant $\varepsilon_2 \in (\kappa\bar{h}, \varepsilon_1)$ exists such that for any $\varepsilon \in (\kappa\bar{h}, \varepsilon_2)$,

$$\frac{d\bar{V}_2(\varsigma_j(t))}{dt} < 0, \forall t \in [t_1, t_2]. \tag{A21}$$

It can be obtained that Equation (A21) contradicts Equation (A1), which means that Lemma 6 is verified. $\square$

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
