# Peer review of "Predictive State Observer-Based Aircraft Distributed Formation Tracking Considering Input Delay and Saturations"

_drones, doi:10.3390/drones8010023_

Round 1
Reviewer 1 Report
Comments and Suggestions for Authors
In this paper, a fully distributed time-varying aircraft formation tracking problem is investigated. The aircraft formation control system consists of an outer loop trajectory control subsystem and an inner loop orientation control subsystem. For aircraft, it is important to consider the time delay of engine response, model uncertainty, tracking capability of the inner loop orientation commands, and other aircraft maneuverability characteristics. To address the input signal delay and model uncertainty, a predictive extended state observer protocol based on fully distributed time-varying formation tracking control (PESO-TVFTC) is proposed. To satisfy the constraints set on the fastness of orientation tracking and smoothness of trajectory tracking, a low-gain feedback technique is introduced into the protocol to keep the control inputs for the outer loop within the desired saturation constraints. Through theoretical analysis, it is proved that multi-aircraft systems can achieve consensus in time-varying formation tracking under certain initial and feasibility conditions, and it is shown that the upper bounds of the PESO gain are limited by the time delay. Numerical simulations are given to demonstrate the effectiveness and improvements of the proposed method.
Comments:
1) It is said (Line 128) that "protocol proposed in this paper is fully distributed". At the same time, it is clear from the paper itself that a classical "leader-followers" scheme is considered, which is not fully distributed. How does this contradiction relate?
2) In the simulation section: qualitative comparison was demonstrated quite well, but there is no quantitative comparison.
3) It is also clear from the simulation section that the authors used a linear model for the simulation. For a linear model, input saturations do not make sense. However, one of the main advantages of the approach is the consideration of input saturations. That is, it is not clear whether the authors performed simulations on such a model, where input saturations affect the flight dynamics itself or not. If not, this should be stated explicitly.
Reviewer 2 Report
Comments and Suggestions for Authors
The paper "Predictive state observer-based aircraft distributed formation tracking considering input delay and saturation" is devoted to development of tracking control algorithm and system stability theoretical proof. The paper can be interesting for formation flying control specialists. There are a set of suggestions and critical comments that can improve the paper.
The main critics is about considered measurement model without measurement noise in eq. (7). The inevitable measurement noise can result in divergence of the proposed control using the same system parameters. To check stability it is recommended to add noise at least at algorithm demostration using simulation example.
It is very difficult to follow the derivatives and proofs because of too many used designation. Some designations are not described in the text (for example, T_0, alpha_0 in (5), u_0 in (8), lambda_0 in (34) etc). Some of the formulas are written with a cut at the edge of the page as at lines 285, 395. The text would be more readable if the authors change the paper structure, put some lemmas and proofs to appendix, leaving only the most essential part in the main text.
The considered in (7) dynamical system according to A, B, C matrices is just a set of free moving particles without dynamics provided in (5) under delayed feedback control. Why do the authors not consider dynamically equation (5) in the PESO design?
In Lemma 7 it is said that all but one eigrnvalues of the Laplasian are located in the left-half plane of the imaginary axis. But later in lines 178-179 there is a statement that L1 has eigenvalues with positive real part, which is in conflict with Lemma 1 statement.
Line 308. What is "syntetic uncertainties"? Please, explain in the text.
Some explanation on choosing such a particular values of the system considered in Section 5 is required.
Reviewer 3 Report
Comments and Suggestions for Authors
This paper presents a state observer based formation tracking control by considering input delay and saturation. First, the graph net is defined, where the leader and follower are also defined. Second, the aircraft model is given. Third, the distributed protocol is designed and the convergence is proved. Finally, the simulation is given to illustrate the effectiveness of the proposed method.
Theoretically speaking, the authors present a distributed control method for a class of formation problem. By considering the input delay and saturation, the proposed algorithm is new.
Technically speaking, the authors use feedback control law and prove that the input is bounded, showing that the saturation can be handled. The design is reasonable.
Please consider the following points when revising the paper
1. For the formation control, the authors may consider the collision avoidance and discuss this issue as suggested in the following references.
(1) J. N. Yasin, S. A. S. Mohamed, M. -H. Haghbayan, J. Heikkonen, H. Tenhunen and J. Plosila, "Unmanned Aerial Vehicles (UAVs): Collision Avoidance Systems and Approaches," in IEEE Access, vol. 8, pp. 105139-105155, 2020, doi: 10.1109/ACCESS.2020.3000064.
(2)Collision avoidance of multi unmanned aerial vehicles: A review,S Huang, RSH Teo, KK Tan,Annual Reviews in Control 48, 147-164,2019
2. From your graph net, I cannot find if all followers can communicate with the leader. Please explain this point.
3. You should notice that the stability analysis cannot guarantee the oscillations of the position formation errors. You may discuss this issue in the paper.
4. In your simulation, can you add more vehicles like ten vehicles?
Round 2
Reviewer 2 Report
Comments and Suggestions for Authors
The authors addressed all the suggestions and questions, the paper can be accepted for the publication
Reviewer 3 Report
Comments and Suggestions for Authors
In this paper, the authors consider a distributed controller based on the estimator. First, the authors introduce the problem statements. Second, the aircraft model is presented. Third, the extended state estimators are designed. Fourth, the stability analysis is given. Finally, simulation studies are given to illustrate the effectiveness of the proposed method.
This is the revised version. I checked the document and find the authors have revised the paper following my comments. I am satisfactory at the current version.